# Impulse-driven oscillations of the near-Earth's magnetosphere

Hiroatsu Sato[1], Hans Pécseli[2,3], Jan Trulsen[4], Per Even Sandholt[3], and Charles Farrugia[5]

[1]DLR Institute for Solar-Terrestrial Physics, D-17235 Neustrelitz, Germany
[2]Department of Physics and Technology, Arctic University of Norway, N-9037 Tromsø, Norway
[3]Department of Physics, University of Oslo, Boks 1029 Blindern, N-0315 Oslo, Norway
[4]Institute of Theoretical Astrophysics, University of Oslo, Boks 1048 Blindern, N-0316 Oslo, Norway
[5]Institute for the Study of Earth, Oceans, and Space, Morse Hall, University of New Hampshire, 8 College Road, Durham, NH, USA

**Correspondence:** Hans L. Pécseli (hans.pecseli@fys.uio.no)

**Abstract.** It is argued that a simple model based on magnetic image arguments suffices to give a convincing insight into both the basic static as well as some transient dynamic properties of the near-Earth's magnetosphere, accounting in particular for damped oscillations being excited in response to impulsive perturbations. The parameter variations of the frequency are given. Qualitative results can be obtained also for heating due to the compression of the radiation belts. The properties of this simple dynamic model for the solar wind – magnetosphere interaction are discussed and compared to observations. In spite of its simplicity, the model gives convincing results concerning the magnitudes of the near-Earth's magnetic and electric fields. The database contains ground based results for magnetic field variation in response to shocks in the solar wind. The observations also include satellite data, here from the two Van Allen satellites.

## 1 Introduction

Instrumented spacecraft in the near-Earth's magnetosphere detect significant dynamic variations in the magnetic fields and plasma properties in response to variations in the solar wind (Araki, 1994; Araki et al., 1997; Archer et al., 2013; Blum et al., 2021). The abrupt increase in pressure associated with interplanetary shocks driven, for instance, by interplanetary coronal mass-ejections (ICME) will compress the low-latitude geomagnetic field through an intensification of the Chapman-Ferraro magnetopause current. This leads to a sudden impulse (SI) which can be observed also in low-latitude magnetometer records. It was demonstrated (Farrugia and Gratton, 2011) that such SI-events are often followed by oscillations of $\sim 5$ min periods. These can be observed also by satellites in the cold, dense magnetosheath and the hot and tenuous magnetospheric plasmas, consistent with also other related observations (Araki et al., 1997; Plaschke et al., 2009). The presence of magnetic pulsations with periods 8-10 min measured by geosynchronous satellites are found to be well correlated with variations in the solar wind dynamic pressure (Kivelson et al., 1984; Sibeck et al., 1989; Korotova and Sibeck, 1995).

A simple dynamic model for the solar wind – magnetosphere interaction was proposed by Børve et al. (2011). In its simplest version the model uses a plane interface between the Earth's magnetic dipole field and an ideally conducting solar wind. This approach has an exact analytical solution in terms of an image method (Chapman and Bartels, 1940; Stratton, 1941; Alfvén, 1950). While the model has tutorial value it is not clear to what extent it can be used for predictions of parameter variations of

the magnetospheric oscillations and overall changes of the magnetosphere in response to abrupt changes in the solar wind. The present study addresses this question using data from space observations obtained using "in situ" data acquired by spacecraft and also ground based observations.

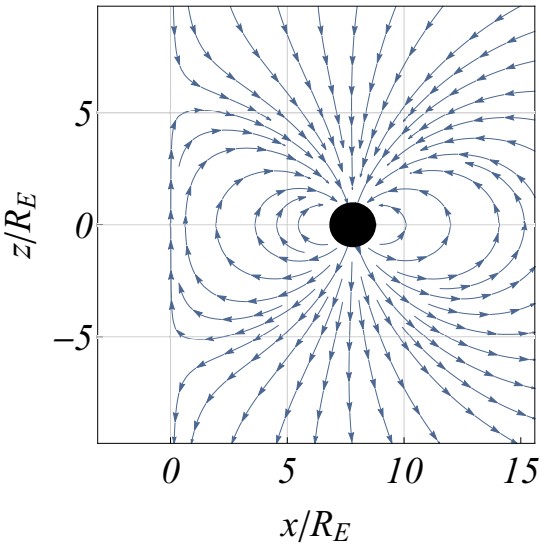

**Figure 1.** *Illustration of a cut in a model magnetosphere assuming a plane interface between the Earth's magnetic dipole field and an ideally conducting solar wind. Distances are normalized by the Earth radius, $R_E$. In this and related following figures, the sun is in the negative x-direction so that x is positive in the direction of shock propagation. The stagnation point of the solar wind is taken at $(x, z) = (0, 0)$. The case illustrated here assumes a strong compression of the magnetosphere by a solar wind pressure pulse by taking the distance to the magnetopause to be $7.8\ R_E$. This value has relevance for data to be shown later. Note the formation of two cusp-points.*

As the impulse from an ICME-shock event arrives at the vicinity of the stagnation point of the solar wind at the magnetopause its perturbation propagates along the magnetosphere with velocity depending on the direction with respect to the magnetic field or the magnetopause. As an order of magnitude we can use

$$\vartheta = \frac{V_A}{\sqrt{1 + (V_A/c)^2}}$$

where $V_A = B/\sqrt{\mu_0 \rho}$ is the Alfvén speed for a plasma mass density $\rho$, and $c$ the speed of light in vacuum. For vacuum or dilute plasmas we have $\vartheta \approx c$, for dense plasmas $\vartheta \approx V_A$. We assume the velocity $\vartheta$ to be sufficiently large to allow the motion of the magnetopause at all relevant points to be assumed nearly instantaneous for the present problem.

## 2   A simple model

In its original form, the basic model (Børve et al., 2011) assumed a plane interface between the solar wind and the near-Earth's magnetosphere. An equilibrium state is found when the solar wind ram-pressure balances the magnetic field pressure at the

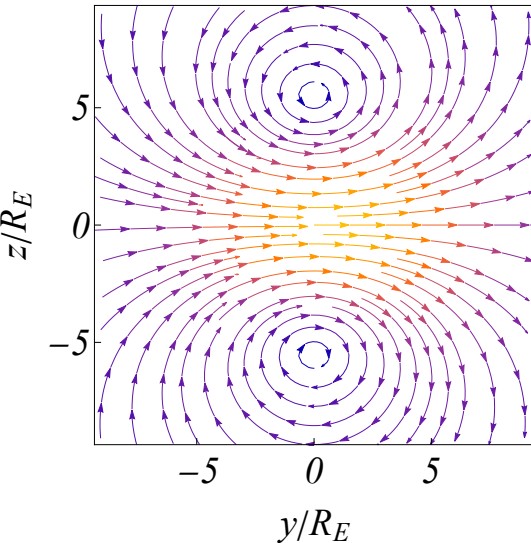

**Figure 2.** *The surface current on the interface between the near-Earth's magnetosphere and the solar wind, consistent with Fig. 1. Note the current loops circling the two cusp points. At equator, the current is directed from dawn to dusk, as obtained by the magnetic field boundary conditions in Fig. 1.*

stagnation point of the solar wind flow as argued by Chapman and Bartels (1940) and Alfvén (1950). Implicit in the argument is that this ram-pressure dominates the electron and ion thermal pressures. The solar wind gives up all its parallel momentum as
in an inelastic collision and flows with a reduced velocity along the interface, i.e. the magnetopause, in a boundary layer with an otherwise unspecified thickness and plasma density. The model predicts static parameters such as the distance between the Earth and the magnetopause (stand-off distance), as well as some dynamic features, in particular the frequency and damping of magnetospheric oscillation in response to an impulsive perturbation in the solar wind. For describing the Earth's magnetic field we here ignore the small tilt of the magnetic axis with respect to the rotation axis. For generalizing the model to other planets
it is straightforward to include such a tilt of the magnetic axis (Børve et al., 2011). The model can be generalized as shown in Appendices A and B. These changes will, however, only have small consequences for the results. In the following we use the simplest version of the model.

### 2.1   Static limit

For the present formulation of the problem the total magnetic field resulting from the Earth's dipole and the Chapman-Ferraro
current can be found by a simple method of images with details as well as figures presented by Børve et al. (2011). The spatial variations of the magnetic field in the near-Earth's magnetosphere predicted by the model are illustrated here in Fig. 1. In particular, the model predicts the distance from the Earth to the stagnation point of the solar wind. The analysis can be generalized to account also for the curvature of the magnetosheath in the vicinity of the stagnation point, see Appendix A.

The surface currents consistent with Fig. 1 are shown in Fig. 2. Near the stagnation point at the magnetic equator the radius of curvature, $\kappa$, increases and $\nabla B$ decreases as compared to the value for a magnetic dipole field in free space.

The equilibrium position $R$ for the stand-off distance from the Earth to the magnetopause is found (Børve et al., 2011) by equating the magnetic field pressure (Ferraro, 1952) from the Earth's magnetic dipole moment $Q_E = 8.0 \times 10^{22}$ A m$^2$ to the solar wind ram-pressure to give the relation

$$R = \left( \frac{\mu_0 Q_E^2}{8\pi^2 U^2 n\overline{M}} \right)^{1/6}, \qquad (1)$$

in SI-units, with $\mu_0 = 4\pi10^{-7}$ H m$^{-1}$ being the permeability of free space, and $n\overline{M}$ is the mass density of the solar wind in terms of number density $n$ and average ion mass $\overline{M}$. The mass density of the solar wind is distinguished from the magnetopause mass density $\rho$. The contribution to the pressure from the weak solar wind magnetic field is assumed to be negligible. A numerical coefficient in (1) is a result of the analysis and not a free adjustable parameter. Expressions similar to (1) can be found in the literature (Walker and Russell, 1995). The scaling with the solar wind dynamic pressure $\left( n\overline{M}U^2 \right)^{-1/6}$ is generally accepted (Southwood and Kivelson, 1990). In fact, apart from a numerical factor, it can be derived from basic dimensional reasoning as shown in Appendix C. The predictions of the model for the distance from the Earth to the magnetopause are shown in Fig. 3 for later reference.

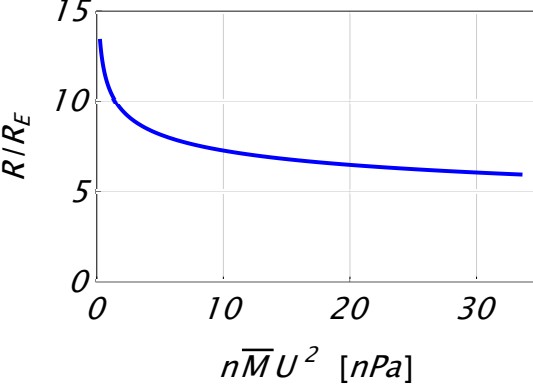

**Figure 3.** *The figure shows model predictions for the stand-off distance from the Earth to the magnetopause (Chapman and Bartels, 1940). The distance is measured in units of the Earth radius $R_E$ shown for varying momentum flux density $n\overline{M}U^2$ (expressed in $nPa$) in the solar wind.*

The surface current that models the Chapman-Ferraro current at the interface between the Earth's magnetosphere and the solar wind at $x = 0$ in Fig. 1 induces a small correction to the magnetic field at the surface of the Earth. A change of the stand-off-distance $R$ in Fig. 3 will give rise to a change in this correction as illustrated in Fig. 4. The illustration assumes a change from a distance of $11\,R_E$ to a new steady state at $7.8\,R_E$. The three spheres show: the absolute value of the change in magnetic field, the absolute value of the change in the horizontal magnetic field component, and finally the change in the normal component with respect to the Earth surface. The latter case is shown including its sign, using the right hand side of

the color-bar. Following the standard convention we have positive values of the vertical magnetic field component $B_v$ pointing into the Earth. The two other cases have units to the left of the color bar. We use this example also to illustrate the effects of a tilt of the magnetic dipole axis, see Fig. 5. For cases relevant for the Earth we find this modification to be of little consequence, and it is ignored in the following. The results shown in Figs. 4 and 5 refer to vacuum fields without including effects of currents in the Earth's near ionosphere.

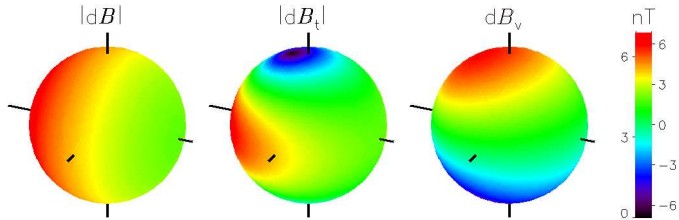

**Figure 4.** *Illustration of the change in the magnetic field, d**B**, in nT at the surface of the Earth in response to a change in the distance between the Chapman-Ferraro current and the Earth. In this case the magnetopause moves from a distance of $11\,R_E$ to $7.8\,R_E$. The sun is to the left with the direction given by a small pointer, used also to give the north-south and east-west directions. The direction of the vertical magnetic field component is positive into the Earth. The 3 figures show $|d\mathbf{B}|$, the absolute value of the tangential component $|d\mathbf{B}_t|$ (left side of the color code), and the vertical component $d\mathbf{B}_V$ (right hand side of the color code), respectively. The change in $d\mathbf{B}_V$ is largest near the magnetic poles.*

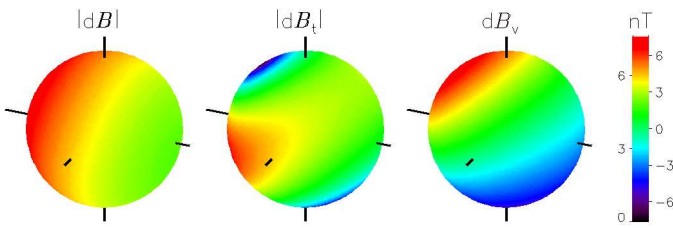

**Figure 5.** *Illustration of the consequences of a tilt of the magnetic dipole axis, here $15°$ in comparison to Fig. 4 where the magnetic dipole axis is vertical. The explanation of symbols is the same for these two figures.*

## 2.2 Dynamic features

In response to an impulse change in the solar wind ram-pressure, the near-Earth's magnetosphere is set into motion. The model of Børve et al. (2011) accounts for this by moving the image dipole in the simple plane interface model as well as in its generalization summarized in Appendix A.

To find oscillating features a physical system needs inertia or its equivalent (Smit, 1968; Cairns and Grabbe, 1994; Freeman et al., 1995). The model is not able to predict this inertia, and it is here quantified by a thickness $D$ and a mass density $\rho$ which has to be determined by observations (Song et al., 1990; Phan and Paschmann, 1996) or numerical simulations that are also available (Spreiter et al., 1966). Analytical models have also been proposed (Cairns and Grabbe, 1994) for the width $D$. Both these parameters vary depending on whether the magnetopause is open (so it has high density solar wind matter inside) or closed. Densities of $\rho = 5 - 25$ cm$^{-3}$ are considered to be typical values. To discuss a finite amplitude nonlinear case, we write Newton's second law for the position of the interface in the form

$$D\rho\frac{d^2}{dt^2}\Delta = n\overline{M}\left(U - \frac{d\Delta}{dt}\right)^2 - 2\frac{\mu_0 Q_E^2}{\left(4\pi\left(R - \Delta\right)^3\right)^2}, \tag{2}$$

where $\Delta(t)$ is the time varying displacement of the interface from its equilibrium value $R$ from (1). The left side of (2) is the product of the mass and the acceleration of a volume element of the moving magnetopause. The first term on the right hand side is the ram-pressure of the solar wind using at any time the relative velocity between the solar wind and the moving interface. The solar wind is assumed to interact with the magnetopause as an inelastic collision. Reflection as in ideal elastic collision does not apply here. The second term on the right hand side is the counteracting magnetic pressure $B^2/\mu_0$ due to the dipolar magnetic field of the Earth taken at the magnetic equator. This force is also derived at the position of the moving magnetopause. We take the sign-convention so that $\Delta > 0$ when the magnetopause boundary moves in the direction of the Earth (this definition differs from the one used by Børve et al. (2011)). The equilibrium solution of (2) with $\Delta = 0$ gives (1).

### 2.2.1 Oscillation frequencies and damping

If we linearize (2) we can derive a scaling law for the characteristic oscillation period as

$$T_0 = \frac{2\pi}{\Omega} = 2\pi\sqrt{\frac{4\pi^2 R^7 D\rho}{3\mu_0 Q^2}} = 2\pi\frac{R}{U}\sqrt{\frac{D\rho}{6n\overline{M}R}}. \tag{3}$$

Apart from the numerical factor, also this result can be found by dimensional reasoning, see Appendix C. A small amplitude damping coefficient can be found as $\sqrt{n\overline{M}U/D\rho}$. Large inertia $\rho D$ gives a long oscillation period and a reduced damping. This is intuitively reasonable since it reduces the velocity of the magnetopause. In a related study (Smit, 1968), a drag force was introduced "ad hoc". Here, the damping is caused by an asymmetry in the solar wind ram-pressure: when the magnetopause is approaching (i.e., moving away from the Earth) the magnetopause is doing work on the solar wind, while in the receding phase it is opposite. The two cases are not symmetric since the effective solar wind ram-pressure depends on the relative velocity between the solar wind and the magnetopause. In the approaching phase this force is large, while it is smaller in the receding phase. The work done in the two oscillation phases is different. Integrated over an oscillation period $2\pi/\Omega$, the oscillations lose energy to the solar wind so the net result is a damping of the oscillations. The initial transient time interval is different: here the solar wind pulse or shock arrives at an interface at rest, and the oscillations are initiated to reach full amplitude.

Magnetopause velocities in the range $10-20$ km/s along the normal to the magnetopause have been reported (Paschmann et al., 1993; Phan and Paschmann, 1996). The velocities depend on plasma parameters, the magnetic shear in particular. The larger of

the values quoted refers to high-shear (Phan and Paschmann, 1996) although it also seems that the observed speeds have a large

statistical scatter. Since the magnetopause is accelerated upon impact from the perturbation in the solar wind (Freeman et al., 1995), these values are only representative, i.e., a large velocity is indicating a large acceleration. The results refer to the satellite frame which is here moving with a velocity much smaller than the magnetopause. Thereby only large magnetopause velocities can be determined unambiguously.

The analysis using (2) and its extensions can readily give also the time variations of the velocity $dZ(t)/dt$ as well as the

acceleration $d^2Z(t)/dt^2$. These results are not shown here since we have no access to relevant data for magnetopause velocities, nor accelerations, for comparison. Concerning the time variation of the position $Z(t)$ we have for comparison indirect results in terms of oscillations in, for instance, the magnetic fields that are induced by the moving magnetopause and thereby the Chapman-Ferraro current systems.

There are alternative and more complicated mechanisms that can give rise to damping, i.e, field aligned currents (FACs) that

flow into the polar ionosphere and further to the global ionosphere, where the energy is consumed by the Pedersen currents (Kikuchi et al., 2021). The damping suggested in the present study is of a different nature.

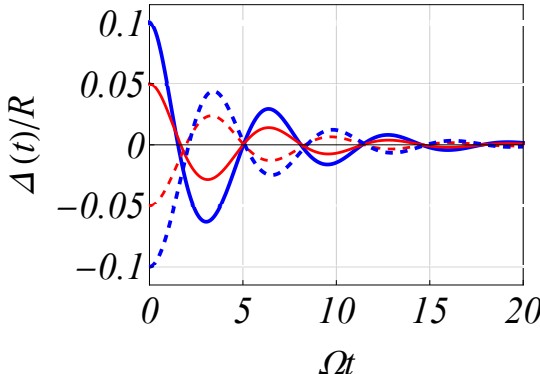

**Figure 6.** *Numerical solutions of the nonlinear normalized equation (4) for 4 pulse-like initial conditions, $\Delta(0)$, two positive corresponding to a compression, and two negative corresponding to a rarefaction. Note that the figure is asymmetric with respect to the $\Delta = 0$ axis.*

The basic dynamic equation (2) can be rewritten in normalized form (Børve et al., 2011) as

$$\frac{d^2}{d\tau^2}Z = -\frac{1}{6}\frac{1}{(1-Z)^6} + \frac{1}{6}\left(1 - \sqrt{\frac{6RnM}{D\rho}}\frac{dZ}{d\tau}\right)^2, \tag{4}$$

with $Z \equiv \Delta/R$ where $R$ is the equilibrium solution (1), while time is normalized by $T_0$ from (3). The basic equation (4)

is strongly nonlinear and the solutions are characterized by significant harmonic generation. Equation (4) can be solved for different conditions, the standard one being where $Z$ is slightly displaced from the equilibrium position to perform damped oscillations, eventually reaching $Z = 0$ as illustrated in Fig. 6. Alternatively, as shown in Fig. 7, we can assume the interface at its equilibrium position until a reference time $\tau = 0$, where there is a sudden and lasting change in the solar wind conditions, changing the equilibrium position. The differential equation has to be modified slightly to account for this case (Børve et al.,

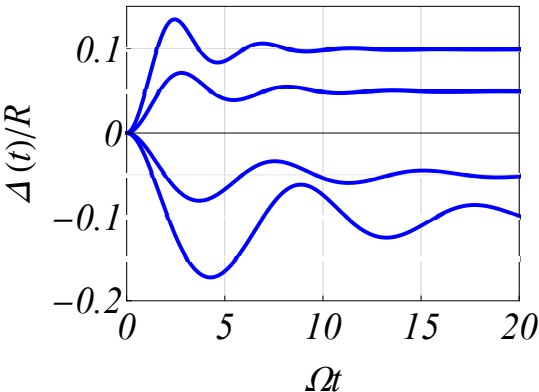

**Figure 7.** *Numerical solutions of the normalized equation (4). For $\Delta > 0$ the solution corresponds to a compression of the magnetosphere in response to a sudden step-like impulse, while $\Delta < 0$ corresponds to a sudden expansion.*

2011). A relevant problem analyzed by the modified version of (2) corresponds to a sudden step-like change in the solar wind plasma density, which we here model by changing $n\overline{M}$ while keeping $U$ constant. The oscillations in Fig. 7 represent transient phenomena occurring between two steady states of the magnetopause, i.e., a time before and one late after the shock arrival. These transient oscillations will modulate those conditions in the Earth's magnetosphere and ionosphere that are induced by changes in the magnetopause.

We estimate the average speed of the magnetopause after it was subject to an impact from a shock-like disturbance in the solar wind by $V = \frac{1}{2}(R_0 - R_1)/T_0$, where its initial position is $R_0$ at time $t = 0$ and the first local maximum the magnetopause displacement is $R_1$ at $t = T_0$, see Fig. 7. We can write this velocity as $V = \frac{1}{2}V_0 \left(1 - (n_1 U_1^2/n_0 U_0^2)^{1/6}\right)$ by using (1) where a representative speed is $V_0 = R_0/T_0$, while the parenthesis is a numerical factor where $\frac{1}{2}$ comes due to the averaging from initial time to $T_0$. Realistic values $T_0 \approx 10$ min and $R_0 - R_1 \approx 4R_E$ give $V \approx 2 \times 6 \times 10^3/6 \times 10^2 = 20$ km/s. This is a large velocity, but it agrees with observations of magnetopause speeds better than an order of magnitude.

The reference calculations in Fig. 7 use $Rn\overline{M} = D\rho/4$, i.e. a relatively large inertia associated with the moving magnetopause. To illustrate the nonlinear character of the oscillations, we show solutions for both positive and negative changes in the solar wind momentum density. For a linear system, the positive and negative parts of Fig. 7 should be mirror images with respect to the horizontal axis. We expect, however, generally a different nonlinear response to a sudden increase and a sudden rarefaction in the solar wind. This may also occur, albeit not as often as compression by a shock. Details of the derivation of the results summarized here are given by Børve et al. (2011), in particular also discussing the simplified linearized limit of the equations.

The physical mechanism causing the damping of the oscillations is found to be an asymmetry in the forcing and the displacement of the magnetospheric boundary. The momentum transfer depends on the solar wind velocity relative to the moving boundary and this is different for an approaching and a receding magnetopause. The damping is thus not due to direct dissipation.

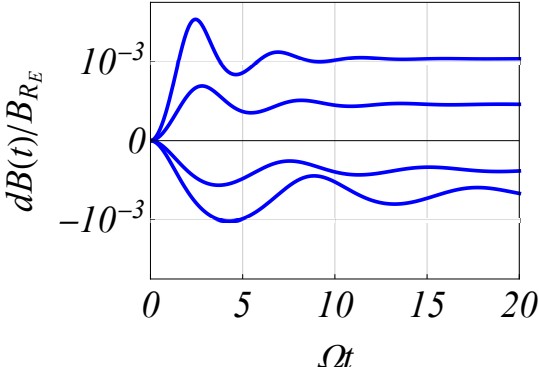

**Figure 8.** *Numerical solutions of the normalized equation (4) used for calculating the variations in the magnetic field as detected at magnetic equator on Earth. The curves correspond to those in Fig. 7. The figure uses $R_E/R = 0.13$.*

### 2.2.2  Time variations of the magnetic fields

The motion of the Chapman-Ferraro current system induces temporal variations in the magnetic field detected at the surface of the Earth. These are illustrated in Fig. 8. The asymptotic limits $t \to \infty$ correspond to Figs. 4 and 5. The analytical expression for the $B$-field perturbation at the magnetic equator as found by use of the image dipole is

$$dB(t) = \frac{\mu_0 Q_E}{4\pi R^3}\left(\frac{1}{(1 - \Delta(t)/R - R_E/2R)^3}\right.$$
$$\left. -\frac{1}{(1 - R_E/2R)^3}\right). \tag{5}$$

The nonlinear features of $\Delta(t)$ are magnified by the analytical form of $dB$, the oscillation period depending, in particular, on the perturbation amplitude as well as its sign. In Fig. 8 we introduced the normalizing by $B_{R_E} \equiv \mu_0 Q_E/(4\pi R_E^3)$ being the $B$-field at magnetic equator at $r = R_E$ for $t = 0$; we have a representative value of $B_{R_E} = 30\ \mu T$. A characteristic value for the perturbation of the magnetic field deduced from Fig. 8 is thus $dB \approx 30\ nT$ for the given parameters.

The time-varying model magnetic field has a straightforward analytical expression in terms of the moving image dipole. The induced electric field can be derived by Faraday's law, as illustrated here in Fig. 9 for the case where the Chapman-Ferraro current system moves with constant velocity. This is here modeled by moving the image magnetic dipole. Note that a calculation starting from the moving Chapman-Ferraro current system would be complicated, while the result for a moving image dipole is simple. Oscillations in $\Delta$ seen in Figs. 6 and 7 give corresponding time variations in the magnetic field. A change in the sign of $\partial \mathbf{B}/\partial t$ gives rise to a corresponding change in the sign of the induced electric field in Fig. 9.

The discussion so far assumed that the density of matter, plasma in particular, is negligible between the Earth and the magnetopause. We discuss next some of the effects on the radiation belts and the Earth's near ionosphere.

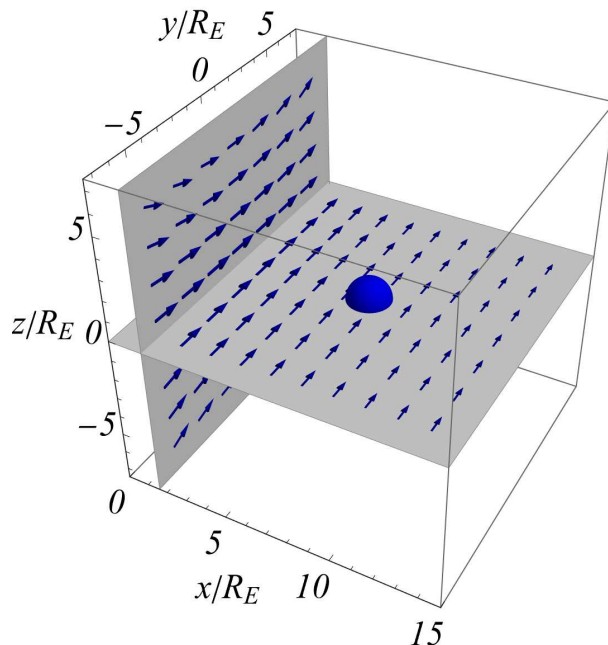

**Figure 9.** *Illustration of the electric field induced by the moving Chapman-Ferraro currents, here modeled by an image dipole (as introduced also in Fig. 1) starting at position $-7.8R_E$ and moving with constant velocity. The direction and relative magnitudes of the electric field are shown with blue arrows. The strength of the electric field reduces strongly at large distances, on the Earth's night side. The position and magnitude of the Earth is shown with a blue sphere.*

### 2.2.3 Motions of the radiation belts.

The moving Chapman-Ferraro current system induces $\mathbf{E} \times \mathbf{B}/B^2$-motions of the magnetic field lines (in the MHD sense (Pécseli, 2012)) in the radiation belts as illustrated in Fig. 10, here shown in the horizontal plane of Fig. 9. Details of individual particle motions are then introduced as corrections to this, e.g. as polarization drifts. In the analysis we assumed initially quiet conditions with the stagnation point at a large distance from the Earth, see Fig. 1, so that the initial boundary of the radiation belts can be assumed circular. The Chapman-Ferraro current system is then allowed to move the stagnation point from $11R_E$ to a distance of $7.8R_E$ from the Earth. We note that the inner boundary is hardly affected since the Earth's magnetic field is too strong there. The deformation of the outer boundary is asymmetric: at the magnetotail side the electric fields are too weak to induce a motion of any significance, see Fig. 9. The outer boundary on the sun-ward side, is on the other hand compressed because the magnetic field is relatively weak while the induced electric field has a sufficient magnitude to give a noticeable $\mathbf{E} \times \mathbf{B}/B^2$-velocity of the magnetic field line motion. The velocity does not matter for a closed adiabatically compressed system: only the initial and final positions of the magnetopause are relevant. For the large magnetopause velocities quoted before (Paschmann et al., 1993; Phan and Paschmann, 1996) we can ignore interactions with the surrounding plasma

and use the adiabatic model. The conclusion is that the moving Chapman-Ferraro current system gives rise to an asymmetric compression of the outer radiation belt, which for the given case amounts to approximately 10%.

The discussion and derivation of the results of Fig. 10 assume that the motion is solely described by the $\mathbf{E} \times \mathbf{B}/B^2$-motion of the radiation belts, with the electric field derived from the motion of the Chapman-Ferraro current system. The sudden impulse-like compression of the radiation belts will act as a "piston" and excite compressional Alfvén waves propagating in the Earth's direction. These waves will give rise to a heating of the plasma that will penetrate deeper into the radiation belts (Zong, 2022). The asymmetry of the day and night sides will however be well represented by results like those shown in Fig. 10.

The sudden accelerated compression illustrated in Fig. 10 gives rise to polarization drifts (Chen, 2016; Pécseli, 2012) of the heavy component, here the plasma ions. For the given geometry, the drift velocity is to a first approximation $\mathbf{U}_D = \Omega_{ci}^{-1} \partial(\mathbf{E}/|B|)/\partial t$, with time varying electric and magnetic fields where $\Omega_{ci}$ is the local ion cyclotron frequency. At the beginning of the shock compression, the associated currents will be in the dusk-dawn direction as described by Araki (1994), and confined to the compressed region. Strong accelerations during the compression give rise to strong currents. Excess charges

will build up at the dawn and dusk boundaries of the compressed regions, see Fig. 11, and these charges can expand only along magnetic field lines or be canceled by ions or electrons flowing up from the ionosphere along the same magnetic field lines. The polarization drift act as generator for these field aligned currents (Araki, 1994). The imposed current density (as modeled by the current generator in Fig. 11) is given as a product of the charge density and the imposed polarization drift, $qnU_D$. The corresponding generator is modeled best by an ideal current generator (Garcia et al., 2015), in contrast to the ideal

voltage or potential generator usually assumed for studies of field aligned currents (Knight, 1973), see also Fig. 11. The ideal current generator has infinite inner impedance while the voltage generator (ideal battery) has vanishing internal impedance (Scott, 1959). It is known (Garcia et al., 2015) that the distinction has important consequences. Numerical simulations of, for instance, ionospheric double layers (Smith, 1982) demonstrated the importance of the generator impedance. Realistic generator models have finite internal resistances and the two generators are related by Thevenin's and Norton's theorems (Scott, 1959).

The potential variations in the circuit, i.e., along magnetic field lines and in the ionosphere, develop in response to the imposed currents. The compression of the radiation belt plasma is modulated by the damped oscillations of the magnetopause. These oscillations in turn modulate the field aligned currents and their time variation will be recognized also in the magnetic fields they give rise to on Earth.

    In response to a change in energy density of the ring current, the Dessler-Parker-Sckopke relations (Dessler and Parker,

1959; Sckopke, 1966) predict a detectable perturbation of the magnetic field as measured by e.g. ground-based stations, but these theorems refer to symmetric conditions. The asymmetric perturbation illustrated in Fig. 10 will take some time to relax and thermalize at a rotational symmetry (Summers et al., 2012), i.e. of the order of 4-6 h for a localized distribution of 3 MeV electrons to transform into a uniformly distributed ring. Some details concerning the dynamics of the radiation belts are summarized in Appendix D.

Shock-induced relativistic electron acceleration in the inner magnetosphere have been observed by instruments on space-crafts (Foster et al., 2015; Tsuji et al., 2017). As the radiation belts are compressed in our model, the plasma will be adiabatically heated (Chandrasekhar, 1960) by a transient process. The effect of the heating will depend on the initial energy of the

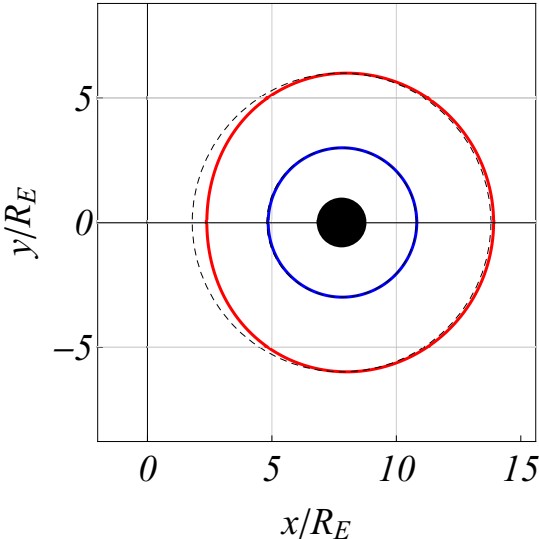

**Figure 10.** *Illustration of the compression of the radiation belts induced by the moving Chapman-Ferraro currents. The figure illustrates the displacement of the inner and outer boundaries. The plane of the figure is perpendicular to the Earth's magnetic dipole. Thin dashed lines show the initial inner and outer boundaries, here taken at positions 3 and 6 in units of Earth radii, $R_E$.*

particles in the radiation belt. Charged particles in the MeV-range will pass through the compressed region in a time that is negligible compared to the compression time and will not be affected. In our model the particle energization is due to the con-
servation of the magnetic moment (Chandrasekhar, 1960) and therefore a bulk plasma heating, the only constraint being that the plasma particles spend more than a few gyro-periods in the compressed region. For protons this time will be approximately $2\pi/\Omega_{ci}(r) = (r/R_E)^3\, 2.2 \times 10^{-3}$ s, taken at the magnetic equator at a distance $r$ from the Earth's center.

The space-time varying electric and magnetic fields generated by the dynamic variations in the position and intensity of the Chapman-Ferraro current system induces currents in the Earth's near ionosphere, the E and F-regions. A simple model for
idealized conditions is outlined in Appendix B.

## 3    Comparison with observations

The model predictions concerning $R$ and $T_0$, as well as the damping of the oscillations received numerical confirmation (Børve et al., 2011). The agreement was even better than stated by the authors due to an incorrect velocity used for normalization in the simulations of their Figs. 9, 10 and 13. In reality the agreement was close to perfect. The numerical model used for
the analysis is however in two spatial dimensions and the steady state conditions depended on numerical resistivity and viscosity that dominate model viscosity and resistivity (Børve et al., 2014). The importance of viscosity is different for numerical simulations in 2 and 3 spatial dimensions. Although this does not affect the dynamic features, a more general test would be worthwhile. Later fully 3 dimensional numerical Magneto Hydrodynamic (MHD) simulations (Desai et al., 2021) have given

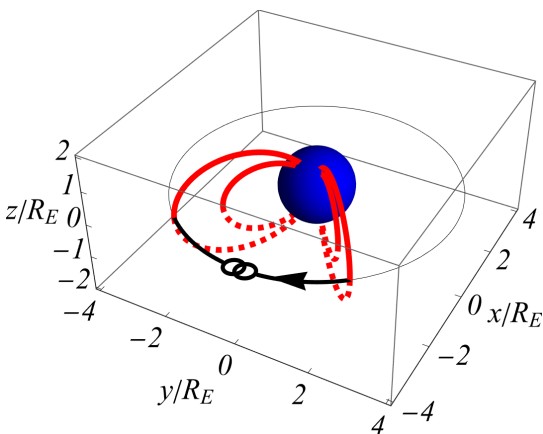

**Figure 11.** *Diagram for illustrating polarization currents (Araki, 1994) generated by an asymmetric compression of the radiation belts, Fig. 10. The current direction is dusk to dawn as indicated by an arrow. Red lines show selected magnetic field lines. The symbol* $-\!\bigcirc\!\bigcirc\!-$ *indicates a current generator. The full circuit is discussed by Araki (1994).*

more detailed results supporting the restricted solutions found by Børve et al. (2014). There is also a slight difference between the 2 and the 3 dimensional versions of the analytical expressions used in the present work.

The predictions of the model discussed in the foregoing can be compared also to space observations as done in the following. Here we distinguish steady state and dynamic observations.

### 3.1 Steady state conditions

Inserting typical numbers as $U \approx 3 \times 10^5$ m/s, $n \approx 5 \times 10^6$ m$^{-3}$, and an average mass equaling the hydrogen mass, $M = 1.66 \times 10^{-27}$ kg, we find $R \approx 7.2 \times 10^7$ m, or $R \approx 11.2\,R_E$ in terms of the Earth radius $R_E = 6.4 \times 10^6$ m. The estimate for $R$ are comfortably within the generally accepted range of $R \sim 10 - 15\,R_E$ (Kivelson and Russell, 1995). The model equation (1) implies a closed scaling law for the distance to the magnetosheet boundary in terms of the solar wind velocity $U$ and the solar wind mass density $n\overline{M}$. Note that there are no free parameters to fit in equation (1), i.e. all are measurable quantities.

### 3.2 Time varying conditions

Solar wind disturbances such as interplanetary (IP) shocks induce significant variations of solar wind parameters during a short time interval, introducing perturbations in the geospace environment, in particular sudden variations of the magnetic fields both in the magnetosphere and on the ground as measured also by ground-based magnetometers (Araki, 1994; Sun et al., 2015). Fluctuations on the minute time scales are often observed in the magnetosphere in response to strong perturbations in the solar wind. Consistent with a model using nonlinear oscillators (Børve et al., 2011), harmonics of the magnetospheric oscillations are often observed (Kepko and Spence, 2003). These are consistent with the strong nonlinear harmonic generation features of the basic model (4). Details of other predictions of the model will here be compared with two sets of observations

| Magnetic field units | nT | nT | nT |
|---|---|---|---|
| Upstream field | -0.13 | 8.24 | -4.02 |
| Downstream field | -4.28 | 13.01 | -10.39 |
| *Velocity units* | km/s | km/s | km/s |
| Upstream velocity | -377.66 | -8.18 | 18.00 |
| Downstream velocity | -447.06 | -35.41 | 22.43 |
| Shock normal | -0.58 | -0.79 | -0.21 |

**Table 1.** *The shock parameters of event December 21, 2014. The vector components are expressed in geocentric solar ecliptic (GSE) coordinates.*

describing responses to shocks in the solar wind. We have chosen two events, December 21, 2014 and March 17, 2015, with significant differences in shock parameters.

### 3.2.1 Event of December 21, 2014.

In Fig. 12 we show plasma and magnetic field data from the *Wind* spacecraft illustrating the propagation of a shock in the solar wind, seen at ∼18:40 UT. *Wind* is located at $\{197.5, -53.5, -8.8\}$ $R_E$ upstream of Earth. From top to bottom, the plot shows the proton density, bulk speed, and temperature, the dynamic pressure, the components of the magnetic field in geocentric solar magnetospheric (GSM) coordinates, and the storm-time $SymH$ index. Parameters relevant to the shock are given in Table 1. The shock is being driven by an ICME (Richardson and Cane, 2010). The magnetic field upstream of the shock (average over

3 min) is $\{-0.13, 8.24, -4.02\}$ nT, and the shock normal, using the magnetic coplanarity theorem (Colburn and Sonett, 1966) is $\{-0.58, -0.79, -0.21\}$, both in GSE coordinates. This gives an angle between the upstream field and the shock normal, $\Theta_{BN} = 52.7°$, so the shock is quasi-perpendicular. The shock speed is $344.6$ km/s, based on Rankine-Hugoniot relations (see Abraham-Shrauner and Yun (1976) and references therein). The Mach number of the shock is ∼4.

In Fig. 13 we show data from the CARISMA magnetometer network in Canada Mann et al. (2008) for a 30 min period. Signals from a few other stations are shown in Fig. 14. An abrupt rise in the magnetic field intensity, followed by some damped oscillations can be seen at about 19:20 UT, where a period of the order of 5–10 min can be noted. The magnitude of the magnetic field perturbations, 10–20 nT are in reasonable agreement with estimates based on Fig. 8. All signals at the 3 stations shown start at the same time, although the stations are widely separated spatially. We have considered a larger number of data

selected in a band around equator and find the same synchronization of the onset with only a small scatter.

   Data from the IMAGE network were also collected, showing somewhat similar results with clear 5 min period oscillations. At the location of the IMAGE stations, the magnetic local time (MLT) at shock arrival is ∼22 MLT, i.e., pre-midnight. At substorm onset the stations are prone to the activation of the westward electrojet. In this case substorm activity was excited by

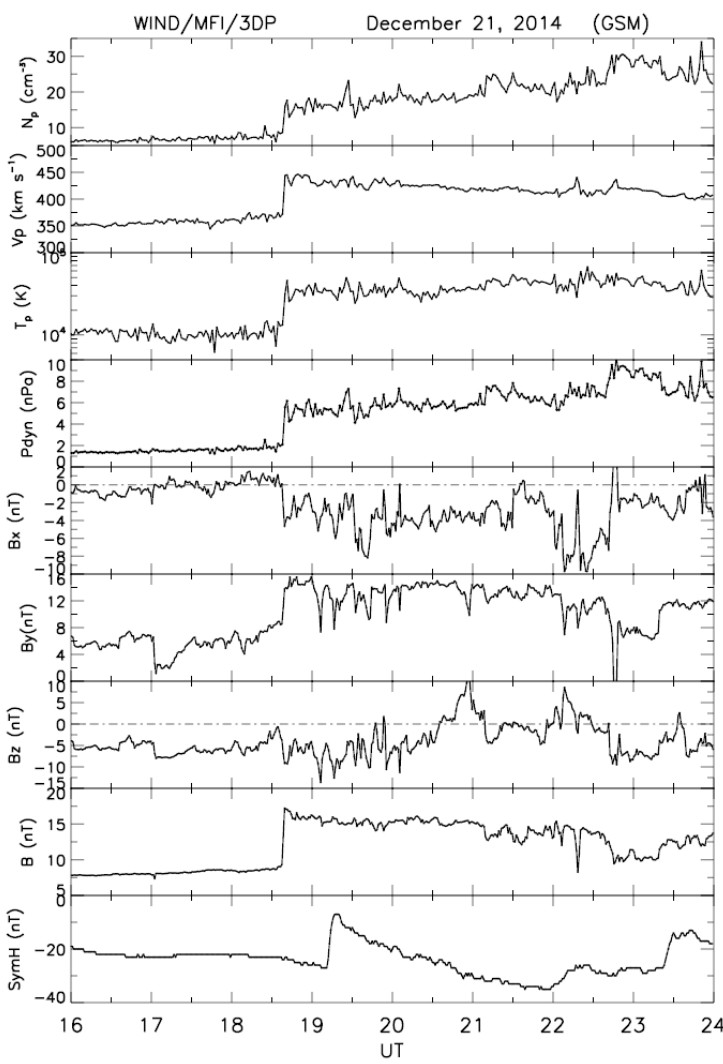

**Figure 12.** *Data from the Wind satellite showing shock propagation in the solar wind to be compared with the results in the following Fig. 16.*

the shock arrival under the prevailing negative interplanetary magnetic field (IMF) $B_z$ conditions (this is seen in the IMAGE magnetograms from the pre-midnight sector; these are not shown here).

The previous figures 13 and 14 were local. To give a more global overview we show the variation of the signal detected by ground-based stations is illustrated in Fig. 15, showing the component normal with respect to ground with a color code. The radius in the small circles give the relative variation of the tangential component of the magnetic field perturbation. An intense magnetic field perturbation with a large vertical component and simultaneously a small horizontal component will thus be shown with a circle having a small radius. The color is red if the vertical component is into the ground, and blue if it is in the opposite direction. The stereographic mapping of the globe is chosen to make the circles having approximately the correct relative magnitudes. When plotting these results we took the first peak maximum after onset of the signal. Note that in general the magnitudes of the horizontal components is larger than the vertical components. We find an overall tendency for positive $B_z$-values in the northern hemisphere and small or negative $B_z$-values in the southern hemisphere. The variation across Northern America appears uniform, in particular. The results are in fair agreement with the model, although not perfect. The strongest deviations are found near the magnetic poles.

### 3.2.2  Event of March 17, 2015.

In Fig. 16 we show data for 1 day (March 17, 2015) from the *Wind* spacecraft, illustrating the propagation of a shock in the solar wind at ∼5 UT. See also Table 2. The field and plasma data are analyzed in the same manner as for previous shock. The magnetic field upstream of the shock is $\{1.75, 4.59, 8.70\}$ nT and the shock normal, using the magnetic coplanarity theorem (Colburn and Sonett, 1966) is $\{0.96, -0.03, -0.28\}$, with both vectors expressed in GSE coordinates. This gives an angle between the upstream field and the shock normal $115.4°$, so the shock is quasi-perpendicular ($\Theta_{BN} = 180° - 115.4° = 64.6°$). The shock speed is 601.3 km/s, based on Rankine-Hugoniot relations, see Abraham-Shrauner and Yun (1976) and references therein. The speed of plasma along shock normal is 405.2 km/s.

In the last panel of Fig. 16 we plot the temporal profile of the $SymH$ index over a 2-day period. $SymH$ is a measure of the strength of the ring current. In this case it has a two-dip structure, indicating that we have a 2-dip storm (Kamide et al., 1998). The weaker dip occurs at ∼9:15 UT, March 17. This caused a major storm already. The $SymH$ then recovers for ∼14 hrs, only to decrease again and reach a new and deeper minimum at ∼21 UT, March 17. The first dip is caused by the shock compressing $B_z < 0$ (GSM) fields. The second one is caused by the long (∼10 hrs) $B_z < 0$ phase inside the ICME itself. Both strengths correspond to major geomagnetic storms, but the second one almost reaches "superstorm" values ($SymH < -250$ nT). The same thing holds for Dec. 21, 2014 event, only the $SymH$ dips are here much weaker (∼35 and ∼60 nT) and they are separated by ∼8 hrs (not shown).

At the time of the sudden impulse seen in the $SymH$ index (at 4:46 UT, March 5) due to the shock seen in Fig. 16, ground stations observed magnetic field variations. Results from the set of the IMAGE magnetometers (Tanskanen, 2009) are shown in Fig. 17. The magnetic local time (MLT) range of the stations in the UT range plotted is 7-9 MLT, i.e., they are sampling

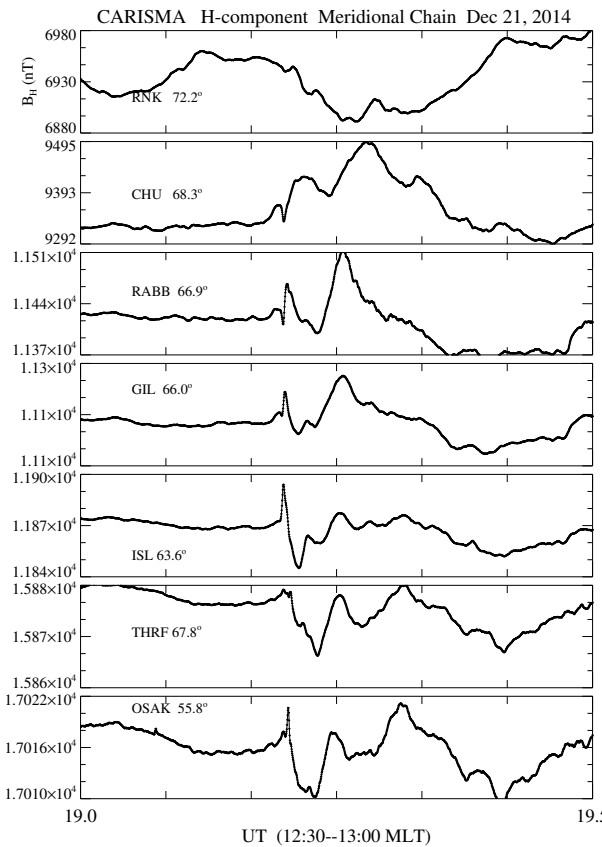

**Figure 13.** *Data from the CARISMA magnetometer network in Canada with the stations specified by their acronyms, RNK, CHU, etc. Apart from the top curve, which refers to an open magnetic field line, we see the pulse arrival followed by low frequency, ∼5 min period oscillations best seen at CGM Latitudes 67.8° and 55.8°. The mean values are not subtracted.*

| *Magnetic field units* | nT | nT | nT |
|---|---|---|---|
| Upstream field | 1.75 | 4.59 | 8.70 |
| Downstream field | 1.98 | 11.63 | 20.52 |
| *Velocity units* | km/s | km/s | km/s |
| Upstream velocity | -421.23 | 27.46 | 2.71 |
| Downstream velocity | -533.99 | 5.44 | 33.34 |
| Shock normal | 0.96 | -0.03 | -0.29 |

**Table 2.** *The shock parameters of event March 17, 2015. The vectors are expressed in geocentric solar ecliptic (GSE) coordinates.*

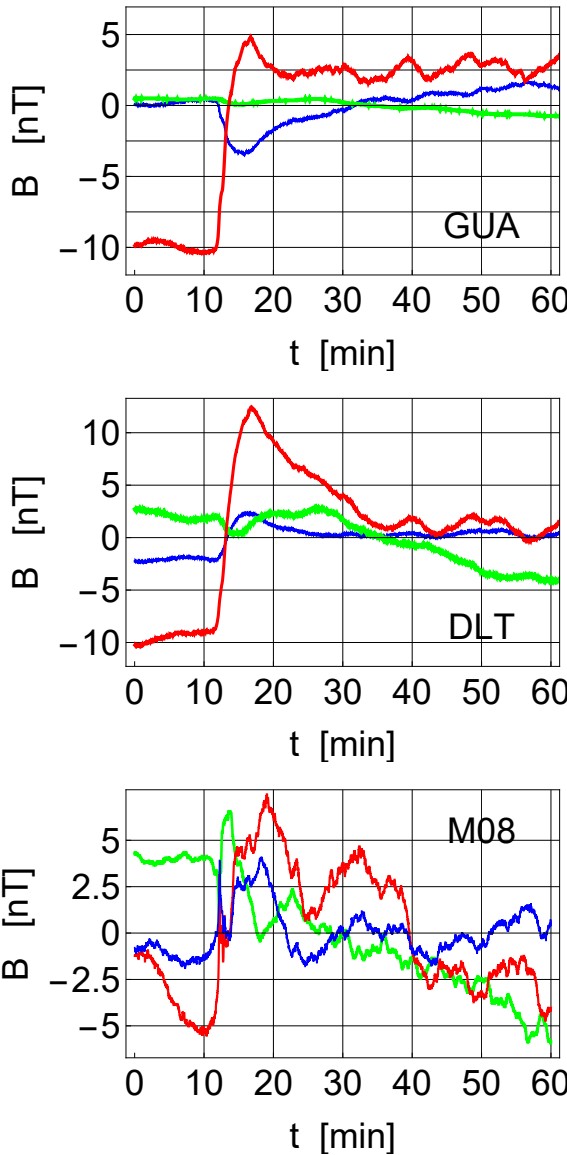

**Figure 14.** *Variations of the magnetic field components $B_x$ (red), $B_y$ (green) and $B_z$ (blue) as observed by the GUA (Guam) and DLT (Dalat, Vietnam) ground stations in response to the shock seen in Fig. 12. Similar data from the M08 (San Antonio) station are shown as well. The averages are subtracted on all figures. The data were obtained by SuperMAG (Gjerloev, 2012). The first data-point is at 2014-12-21, 19:00 UTC. We note some heavily damped oscillations in all figures, where oscillations with 5 - 10 min periods are discerned.*

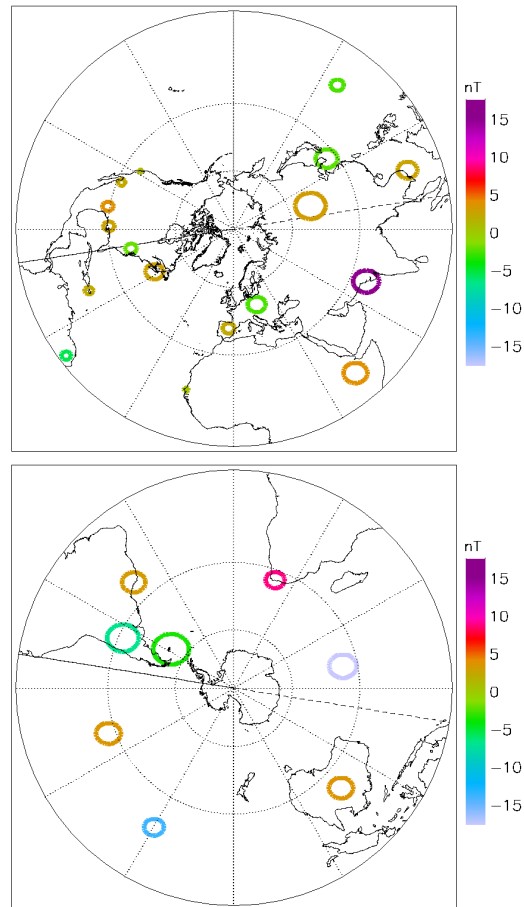

**Figure 15.** *Variations of the magnetic field components as detected on ground on 2014-12-21. The data refer to the first peak value after the shock arrival in figures like Fig. 14. North and South America are facing the Sun at this time. The data were obtained by SuperMAG (Gjerloev, 2012). The mapping of the circles indicating the horizontal* B*-component follow mapping the continents.*

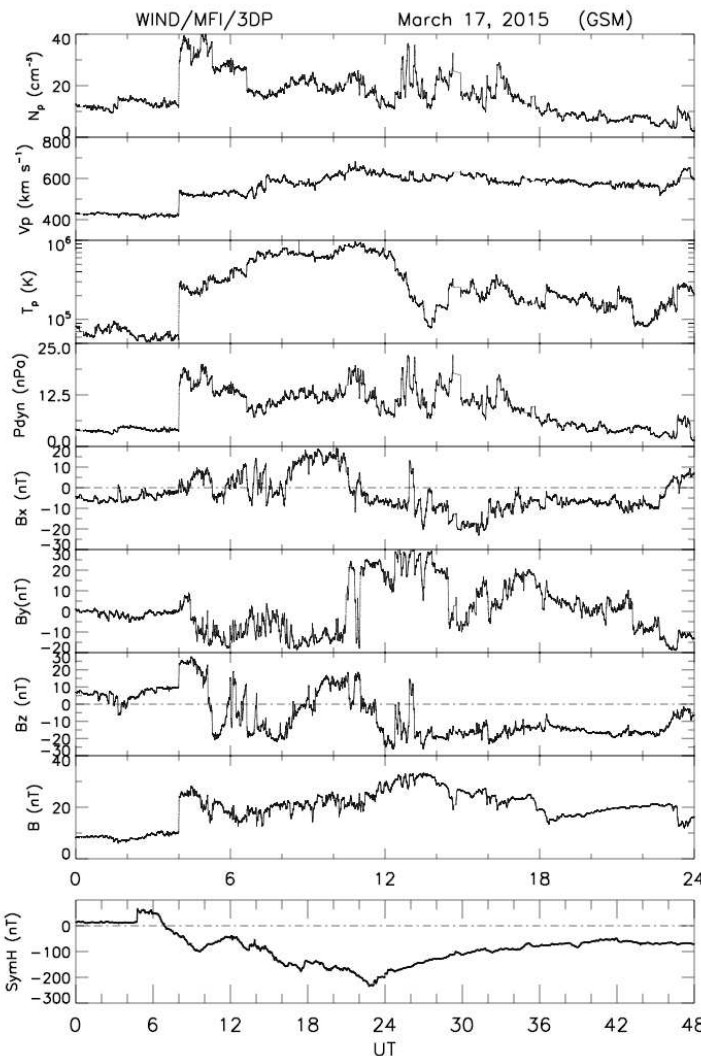

**Figure 16.** The figure shows *Wind* satellite and ground-based data for the period March 17-18, 2015. The format is the same as Fig. 12, except that the proton beta in the last panel of Fig. 12 has been replaced by the storm time $SymH$ index. At ∼4:03 UT, a sharp rise is seen in $Np$, $p$, $Tp$, $Pdyn$ and $B$, indicative of a shock passing the *Wind* spacecraft. The ground response is recorded ∼ 43 min later with a rise in the index $SymH$. The configuration resulted in a geomagnetic storm reaching peak values of $-234$ nT. For completeness and later reference we show $SymH$ results separately for a duration of 48 h (17-18 March, 2015).

dawn-side local times. Figure 17 shows magnetograms from a wide range of latitudes extending from the polar cap to middle latitudes. The negative/positive deflections at the northernmost stations on Svalbard (NAL to BJN) are related to the activation of lobe cell convection under the strongly northward interplanetary magnetic field (IMF) condition at the time of the shock arrival (see Fig. 16).

The bipolar signal seen seen in our Fig. 17 at auroral latitudes corresponds to the DP-type perturbations described by Araki (1994). The interpretation of this signature given (auroral zone, morning local time) is in terms of an M-I-coupling illustrated in Fig. 12 in that work. The signatures described shown in the present work at Svalbard latitudes are explained in terms of lobe-cell polar cap convection with an associated Hall-current.

Our focus in the present study is on the impulse/oscillation at lower latitudes during the interval 04:46-04:50 UT, which is more directly related to the IP shock. After a short transition we see small-amplitude, damped oscillations of periods 4-6 min at 4:46 UT. The signal obtained by selected ground stations at various local times is illustrated in Fig. 18. Here the MLTs are: at GUA, 14:30 MLT, at DLT, about 12 MLT and at M08 about 22 MLT. Note the vertical scales are larger than those of Fig. 14, consistent with the shock intensity. From Figs. 12 and 16 we estimate the solar wind pressure in the event of March 17-18, 2015 to be approximately twice as large as in the event of December 21, 2014, implying that the characteristic frequency $\Omega$ in the former case is $\sim \sqrt{2}$ larger than in the latter case. The oscillation period is readily estimated visually in Figs. 14 and 18 but the corresponding local frequencies can also be estimated by a wavelet transform (Kaiser, 1994). Illustrative results are shown in Fig. 19. A full wavelet analysis of the signals from a large representative dataset fall outside the scope of the present study, but we note that the wavelets reveal oscillations of $0.5 - 2$ mHz trailing the step-like magnetic field enhancement originating from the solar wind shock. This numerical value agrees with our model.

The signal obtained by ground stations at various local times is illustrated in Fig. 20. As in Fig. 15, we show the component normal with respect to ground with a color code. Positive values point into the Earth also here. The results near the magnetic poles have magnitudes typically up to 2–3 times larger than the average of the values shown, and also the time variations found there can be more irregular. The same comments apply also to Fig. 15 and the values for these regions are not shown. These polar features are believed to be due to field aligned currents (Knight, 1973; Lühr and Kervalishvili, 2021), not accounted for in the present model. Ground magnetic disturbances at auroral and sub-auroral latitudes can be induced by both ionospheric and magnetospheric currents (Araki et al., 1997). At middle and low latitudes, the cause of magnetic field disturbances are dominated by magnetopause currents superimposed by weak ionospheric currents. At the dayside equator, the ionospheric Cowling currents are the major source for the equatorial Sudden Commencements (SC). The spatial variations in the magnetic field perturbations seen in Fig. 20 are larger and more non-uniform compared to those found in Fig. 15. We take this as an indication of a stronger influence of the ionospheric currents in the latter case where the solar wind shock is strongest.

For the dynamics in the radiation belts we have data from the Van Allen Probes (formerly known as the Radiation Belt Storm Probes (RBSP)), with the relevant positions of the two satellites deep inside the magnetosphere shown in Fig. 21. The satellites measure the electric fields as shown in Fig. 22. From (1) we have for this case $R \approx 6 \times 10^7$ m, and from (3) we find $T_0 \approx 250$ s, or approximately 4 min. The magnetopause moves approximately $0.05R$ within a time $T_0/5$, giving a velocity $0.25R/T_0 \approx 6 \times 10^4$ m s$^{-1}$ or 60 km s$^{-1}$ using the estimate from Fig. 7. To represent the temporally changing magnetic field,

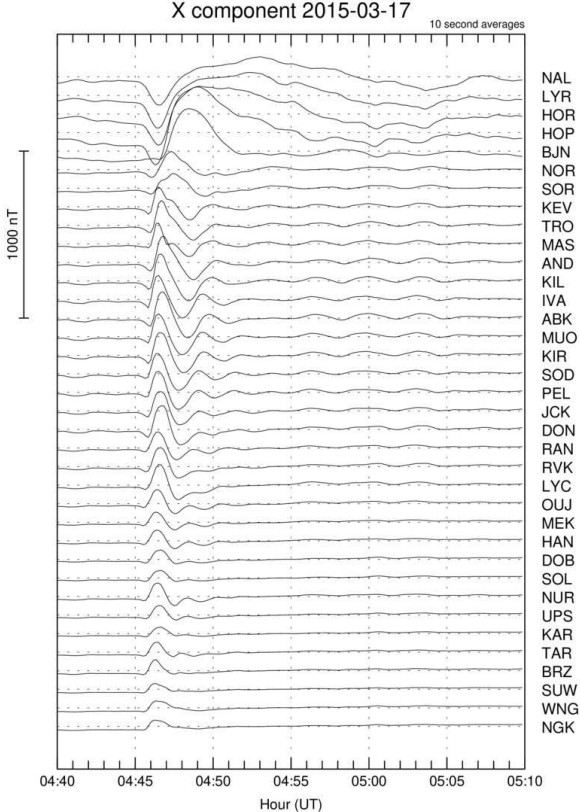

**Figure 17.** *X-component of the signals detected at the IMAGE stations at the event of March 17, 2015. Each curve is labeled by the acronym for the appropriate station. The data are obtained near 04:46 UT, i.e., around 07:46 MLT (magnetic local time), near dawn. The questions relating to the substorm current wedge mentioned earlier do not apply for this case.*

the image dipole has to move with a velocity $U$ twice this value. The magnetic field from the moving image dipole has to be taken at a distance of approximately $1.5\,R$ from it, i.e. at a position somewhere between the Earth and the magnetopause, giving $B \approx 20$ nT. With $E \approx UB$ we estimate $E \approx 2$ mV m$^{-1}$ in the negative $\hat{\mathbf{y}}$-direction. This is a value derived for vacuum conditions, while the two probes are located inside the radiation belts, subject to the dielectric plasma shielding. The dominant component of the detected electric field value is in the positive $\hat{\mathbf{y}}$-direction, see Fig. 22. We note heavily damped electric field oscillations with approximately 2 min period. The two satellites (both on the night side) are at similar distances from the magnetopause and detect similar electric fields. They are however placed at different positions in the radiation belts so the observed particle energy variations are different. In the present context, the electric field measurement act as a marker for estimating the time-delay of the plasma heating pulse.

The time variation of the energy distribution of the plasma in the Earth's radiation belt was measured as shown in Figs. 22, 23 and 24. It is the Van Allen Probe $A$ that detects the strongest electron heating in Fig. 22. The REPT and magEIS instruments

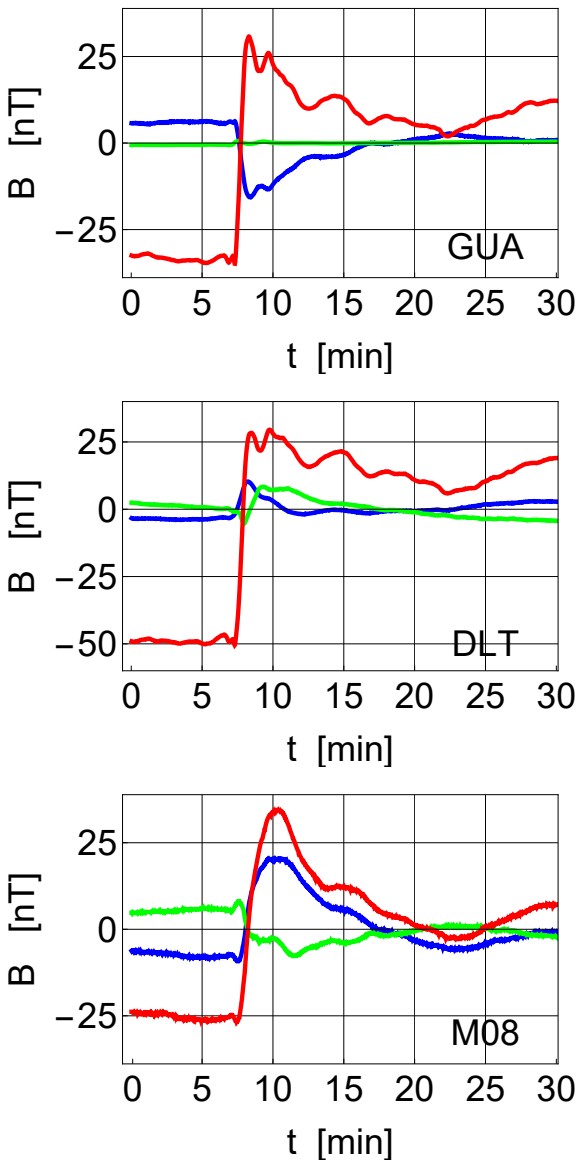

**Figure 18.** *Variations of the geomagnetic field components, $B_x$ (red), $B_y$ (green) and $B_z$ (blue), as observed by ground stations GUA (GUAM), DLT (Dalat) and M08 (San Antonio) in response to the shock seen by Wind (Fig. 16). The magnetic local times being sampled are 4 MLT (GUA), 3 MLT (DLT) and 12 MLT (M08). The averages are subtracted in all figures. The first data-point is at 2015-03-17, 04.38 UTC. Notice the damped $\sim 5$ min period oscillations in the figures. See also Fig. 14 for comparison.*

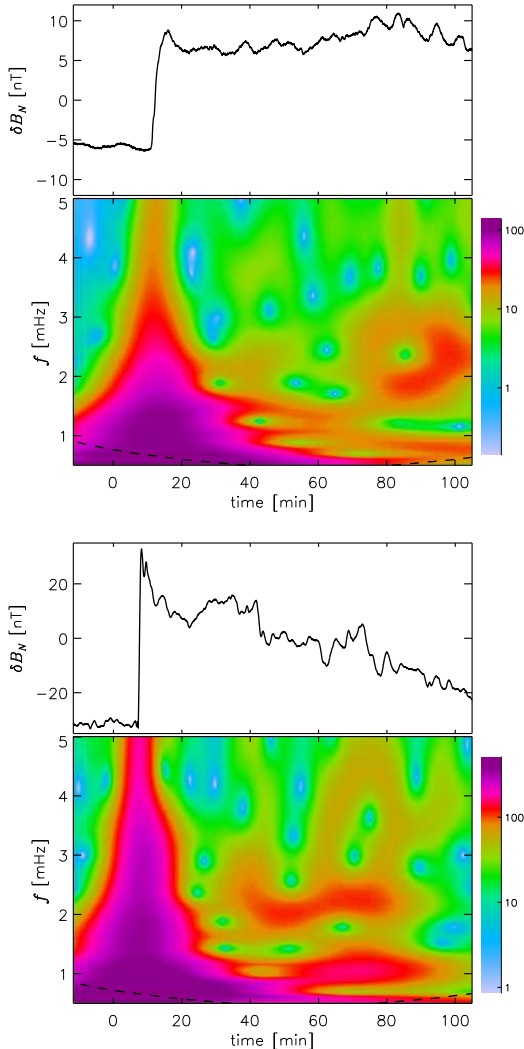

**Figure 19.** *Time-frequency representation using a Morelet-wavelet for the magnetic X-component of the signal from the GUA-station, see Figs. 14 and 18 for 2014-12-21 and 2015-03-17, respectively. Note the harmonic content in both samples. The "trumpet-like" form in both figures is the wavelet transform of a step discontinuity originating from the shock impulse. The signals are affected by "edge effects" for frequencies below the dashed line. The results shown here are representative for other stations.*

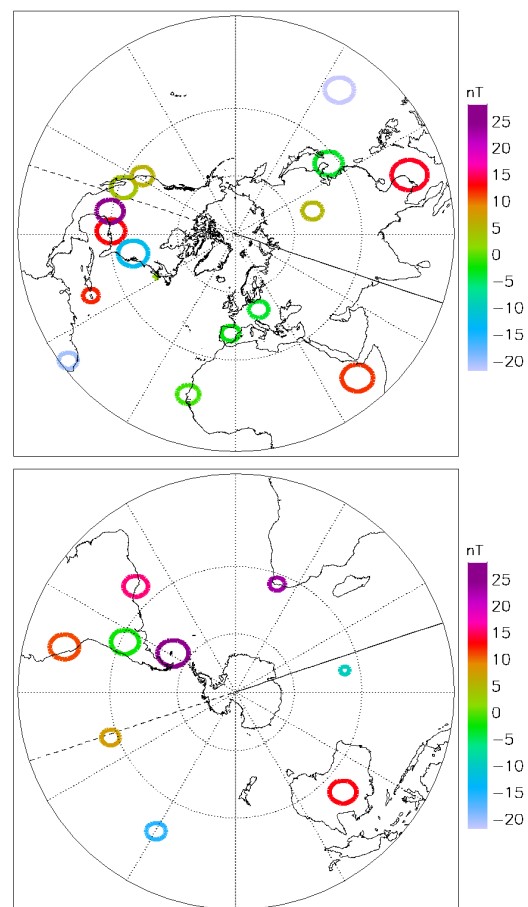

**Figure 20.** *Variations of the magnetic field components as detected on ground on 2015-03-17. The data refer to the first peak value after the shock arrival in figures like Fig. 18. The data were obtained by SuperMAG (Gjerloev, 2012).*

(Blake et al., 2013; Baker et al., 2013) on probe $B$ show particle heating at a somewhat smaller level compared to probe $A$. The heating signal arrives a little earlier (by approximately 1 min) at probe $B$, see Fig. 21 for the probe positioning. The electrons energized by the compression of the radiation belts as shown in Fig. 10 will have their $\mathbf{B} \times \nabla B$-drift in the positive $y$-direction in Fig. 21. These electrons have to propagate $\sim (4/3)\pi R_E$ to reach satellite $A$. The estimate in Appendix D is in reasonable agreement with the time delay found in Fig. 22. Due to the compression of the sunward part of the Earth's magnetic field, the estimate of charged particle velocities based on a magnetic dipolar field as in Appendix D will only serve as a guideline.

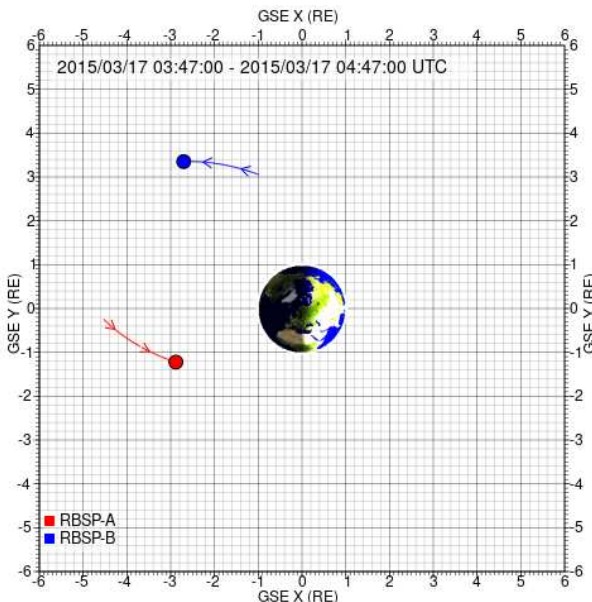

**Figure 21.** *Positioning of the two Van Allen satellites with $A$ at approximately the position $(-2.8, -1.2)$ and $B$ at $(-2.7, 3.3)$. The orbits are close to being confined to the $x-y$-plane, so only the projection of the orbits on this plane is shown. The satellite positions are shown at time* 04.46 *UTC. The duration of the trajectory shown is 1 h. The Sun is to the right in this presentation.*

## 4   Conclusions

A simple model for illustrating the near-Earth magnetospheric static as well as dynamic features has been presented. The model predicts the distance between the Earth and the magnetopause (stand-off distance) without introducing free parameters. Some dynamic features, in particular the frequency of magnetospheric oscillations in response to an impulse in the solar wind is derived as well. The parameter variations of the oscillation frequency is expressed analytically. A damping of the oscillations is predicted, in particular also its variation with solar wind parameters. This damping is not caused by dissipation but is an
inherent feature of phase relations in the model. For testing the predictions of the model we considered two events, i.e., two geomagnetic storms: one moderate for the December 21, 2014 with $SymH \sim -70$ nT and a strong one occurring at March 17, 2015 with $SymH \sim -237$ nT. The magnetospheric response on the impact of them was similar, but with significant differences in the details. The agreement with our model was best for the moderate shock. The magnetic field perturbations in Fig. 18 are significantly larger than those shown in Fig. 14 as expected for the stronger shock. The observed oscillations are consistent
with results reported by other studies (Plaschke et al., 2009; Farrugia and Gratton, 2011). Considering the simplicity of our model, we find its overall agreement with observations to be satisfactory. The basic ideas apply for other magnetized planets as well.

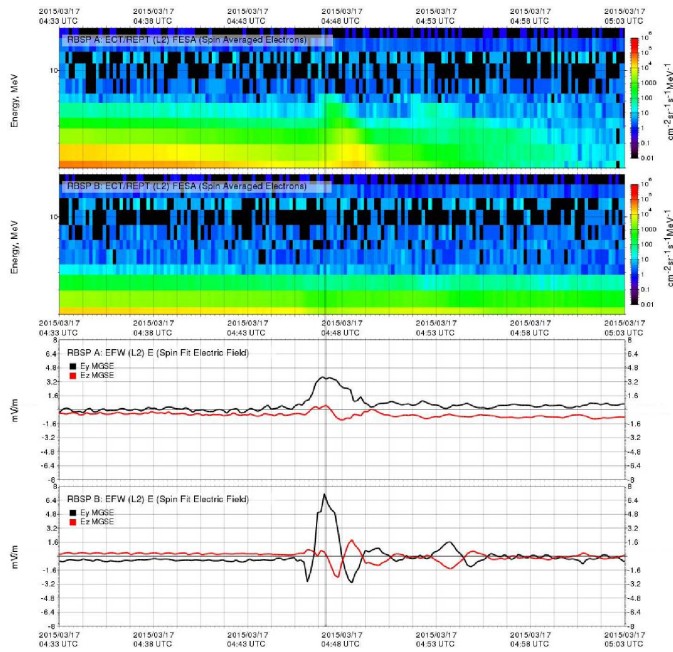

**Figure 22.** *Temporal variation of the electron energy distribution (top) and electric field components (bottom) as detected by the two Van Allen Probes. A thin vertical gray reference line indicates the arrival time of the electromagnetic pulse at the Van Allen satellites. Note the time-lag of the detected electron heating. The lowest energy particles are delayed most. For 2 MeV particles the delay is approximately 50 s as found on satellite A. There is only negligible electron energization detected by satellite B. Strongly damped oscillation in the electric fields have a period of approximately 2 min.*

The main conclusion of the present study can be summarized in few words: "Three dipoles suffice for the lowest order modeling of the near-Earth magnetosphere", one, $\mathbf{Q}_E$, for the Earth's magnetic field and two image dipoles, where one, $\mathbf{Q}_I$,

is placed in the solar wind, the other, $\mathbf{Q}_S$, in the Earth's interior see Appendix B. We have $\mathbf{Q}_I$ and $\mathbf{Q}_E$ to be parallel so their magnetic field contributions add at the Earth's surface. The interior dipole $\mathbf{Q}_S$ is anti-parallel to $\mathbf{Q}_E$ so that the radial component of the magnetic field cancels in the the ionosphere at a radius here taken to be $R_E$ to sufficient accuracy.

One partial result of the present analysis is an emphasis of the strongly nonlinear features and damping of the magnetospheric oscillations. These are explained here by the basic properties of a simple physical model. The observed frequencies and

damping rates seen in, e.g., the CARISMA data in Fig. 13 and in part also the IMAGE data in Fig. 17 are in good agreement with the model results. Ground-based results by SuperMAG are in similarly fair agreement, see Fig. 14 and also Fig. 18. By inspection of Figs. 12 and 16 we find that there are no systematic long period oscillations of $|\mathbf{B}|$ following the shock structure. The oscillations observed in figures like 13 or 18 are thus natural for the system and not due to some external forcing. The model also predicts the magnitudes of magnetic and electric fields detected by ground stations and satellites to better than

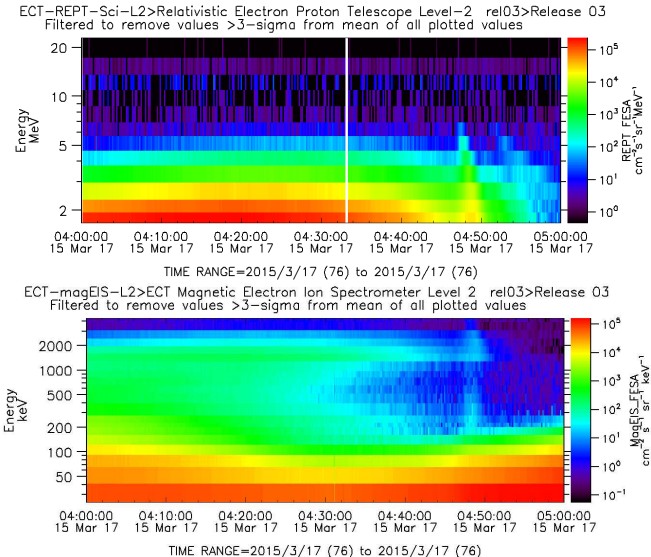

**Figure 23.** *Temporal variation of the energy distribution of the plasma particles forming the radiation belts as detected by the Van Allen Probe A. Two energy resolutions of the same event are shown, illustrating that it is the lowest energy particles that are heated most. Note the dispersion in particle velocities: the most energetic particles arrive first, see the top frame. Data from the REPT and magEIS instruments are shown. The vertical white line is a data-gap.*

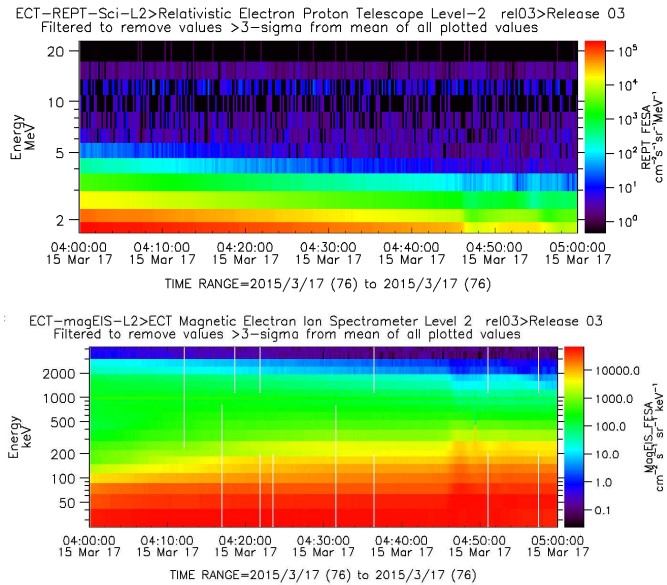

**Figure 24.** *Temporal variation of the energy distribution of the plasma particles forming the radiation belts as detected by the Van Allen Probe B. See also Fig. 23. Some thin white vertical lines are data-gaps.*

an order of magnitude. We consider this to be satisfactory. The ideas put forward in the present study can be applied to any magnetized planet like the Earth, orbiting a star like the Sun.

The main limitations of the model are found in:

- its inability to account for the far magnetotail conditions, the dynamics in particular. A cross-tail current is not included in the model.

- Magnetic field line reconnection (Califano et al., 2009) is missing in the model. This is important when the interplanetary magnetic field has a downward component. Field aligned currents (FAC) following reconnection are consequently not accounted for.

- Surface eigenmodes of the dayside magnetopause (Hwang, 2015; Hartinger et al., 2015; Archer et al., 2019) are not accounted for. These will give rise to additional, presumably small amplitude, oscillations to the modes described in our 395 work.

- The model gives only a schematic account for the excitation of Alfvénic waves and the particle accelerations associated with them.

We believe the present simple model deserves scrutiny. The predictions can be compared to other related data which can be classified according to details such as $SymH$ for the observed shocks.

*Acknowledgements.* Valuable communications with Jesper Gjørlev concerning use and interpretation of SuperMAG data are gratefully acknowledged. Magnar Gullikstad Johnsen offered valuable comments. One of the authors (HLP) thank Gérard Chanteur for valuable comments. CJF was supported by NASA Wind grant 80NSSC19K1293 and 80NSSC20K0197. Figure 14 presented in this paper rely on the data collected at the GUA, DLT, and M08 stations. We thank Institut de Physique du Globe de Paris and United States Geological Survey for supporting their operation and INTERMAGNET for promoting high standards of magnetic observatory practice (www.intermagnet.org). We 405 thank the institutes who maintain the IMAGE Magnetometer Arrayand also I.R. Mann, D.K. Milling and the rest of the CARISMA team for data. CARISMA is operated by the University of Alberta, funded by the Canadian Space Agency. Spacecraft Wind data at 3 sec resolution were acquired by the Wind Magnetic Field investigation (MFI; Lepping et al. (1995)) and the Wind 3D Plasma Analyzer (3DP; Lin et al. (1995)). They were downloaded from NASA's CDA website. Data from the Van Allen probes were accessed at the Science Gateway, maintained by the Johns Hopkins University Applied Physics Laboratory. We thank Bern Blake, Joe Fennell, Seth Claudepierre, and Drew Turner 410 for use of the MagEIS data and Dan Baker, Shri Kanekal, and Alyson Jaynes for use of the REPT data. Processing and analysis of the [HOPE, MagEIS, REPT, or ECT] data was supported by Energetic Particle, Composition, and Thermal Plasma (RBSP-ECT) investigation funded under NASA's Prime contract no. NAS5-01072. All RBSP-ECT data are publicly available at the Web site http://www.RBSP-ect.lanl.gov/

*Author contributions.* All authors contributed to the data collection, analysis and manuscript preparations. HS, PES, and CJF mostly to the data collection, HLP and JKT mostly to the analysis and the preparation of the manuscript and figures.

*Competing interests.*  The authors declare that they have no conflict of interest.

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

## Appendix A:  Generalization of the model

In empty space we approximate the Earth's magnetic field by a simple dipole, here written in spherical coordinates

$$B_\phi = 0, \quad B_\theta = \mu_0 Q_E \frac{\cos\lambda}{4\pi r^3}, \quad B_r = \mu_0 Q_E \frac{2\sin\lambda}{4\pi r^3}, \tag{A1}$$

in terms of the angle $\lambda$ measured from the magnetic equator. We introduced the magnetic dipole moment $Q$. For the Earth we have $Q_E \approx 8 \times 10^{22} \, \mathrm{A\,m^2}$.

Assume that the cut in interface between the solar wind and the Earth's magnetosphere can be approximated locally by a
circle with radius $R_0$, see Fig. A1. The angle between $\mathbf{R}_0$ and the line connecting the Earth and the Sun is $\Psi$. At an angular position $(r, \lambda)$ on the interface, see Fig. A1, we can require (at least for small $\Psi$, away from the cusp-points) an approximate balance in the form

$$\overline{M}n(U\cos\Psi)^2 \approx \frac{1}{2\mu_0} B^2(r, \lambda). \tag{A2}$$

Expression (A2) states that the normal component of the solar wind "RAM"-pressure balances the magnetic field pressure,
keeping in mind that the magnetic field lines are parallel to the curved interface. The magnetic field pressure decreases in the $z$-direction away from the stagnation point, and the component of the solar wind velocity normal to the interface decreases for increasing $\Psi$ as well. The magnetic field on the Earth-ward side of the interface results from the sum of the $B$-fields from the Earth's magnetic dipole (A1) and an image dipole. Due to the manageable boundary conditions for potentials and electric

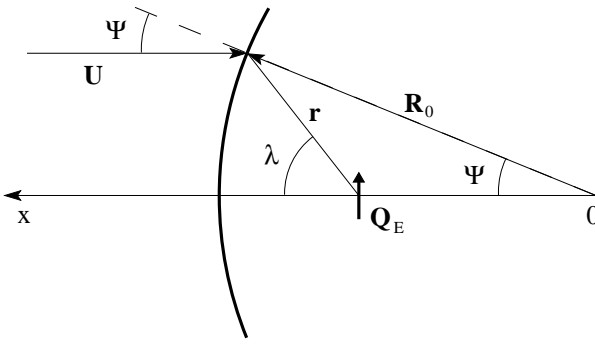

**Figure A1.** *Illustration of coordinate system for a modified model of the Magnetospheric interface with $\mathbf{Q}_E$ being an equivalent dipole for the Earth's magnetic field. The interface follows the magnetic field lines for small $\Psi$.*

fields, the method of images is relatively simple for electric point charges, dipoles, etc., in the vicinity of ideally conducting surfaces. For magnetic dipoles, and higher order multi-poles, the corresponding problems are simple only for some special cases (Ferraro, 1952; Spreiter and Summers, 1965), a plane boundary being one of them.

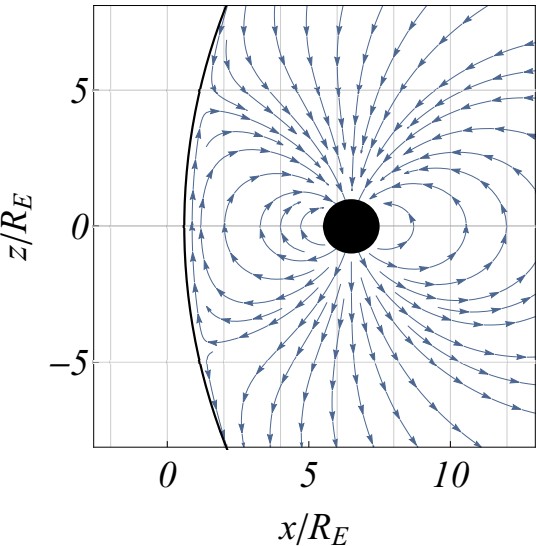

**Figure A2.** *Illustration of a generalization of the results from Fig. 1 where the image method is extended to account for a self-consistent curvature of the interface between the solar wind and the Earth's magnetic field.*

To obtain an approximation for the magnetic field between the Earth and a curved magnetosphere (Spreiter and Summers, 1965) we here take two parallel dipoles, the Earth's magnetic dipole $\mathbf{Q}_E$ and an image dipole $\mathbf{Q}_I$ at positions $x_E$ and $x_I$, respectively. Introducing the magnetic field from a dipole (A1) and the given radius of curvature for the ideally conducting

surface we find that to lowest approximation $\mathbf{Q}_I \approx \mathbf{Q}_E(R_0/x_E)^3$ placed on the $\widehat{\mathbf{x}}$-axis at the position $x_I \approx (R_0/x_E)^2 x_E$. Near the stagnation point, the total magnetic field becomes

$$B(\mathbf{r}) = \frac{\mu_0 Q_E}{4\pi} \left( \frac{1}{|\mathbf{r} - x_E \widehat{\mathbf{x}}|^3} \right.$$
$$\left. + \left( \frac{R_0}{x_E} \right)^3 \frac{1}{|\mathbf{r} - x_E(R_0/x_E)^2 \widehat{\mathbf{x}}|^3} \right), \tag{A3}$$

a result that can be derived from the related problem for an electric dipole near a conducting sphere. The first term gives the Earth's magnetic field, the second is the image field. With the notation of Fig. A1, we have $r = \sqrt{R_0^2 + x_E^2 - 2R_0 x_E \cos \Psi}$ and $\sin \lambda = (R_0/r) \sin \Psi$. Introducing (A3) in (A2) it is convenient to define a characteristic scale $C_L \equiv 2(4\pi)^2 \overline{M} n U^2 / (\mu_0 Q_E^2)$ having the physical dimension $length^6$. The origin of the coordinate system is not specified but here determined through $R_0$ and $x_E$. We could choose to place the origin at the Earth, but the analytical expressions will become more complicated, and we would then have to determine $R_0$ as well as the position for the radius of curvature. For the stagnation point (stand-off-distance) of the solar wind at $(x, z) = (R_0, 0)$ the expression (A3) with (A2) gives the relation $n\overline{M}U^2 = B^2(R_0, 0)/2\mu_0 = 2\mu_0 Q_E^2 (4\pi)^{-2}(x_E - R_0)^{-6}$, in particular. This is consistent with the result of Børve et al. (2011), since $R_0 - x_E$ is the distance between the Earth and the interface between the solar wind and the magnetopause. A plane interface is a good approximation at $(R_0, 0)$. Given the parameters we use (A2) to determine the radius $R_0$ that eliminates the $\Psi$ dependence, at least to lowest approximation. A stronger solar wind pressure gives a smaller radius of curvature. As the solar wind pressure decreases, i.e. $n\overline{M}U^2 \to 0$, we have $C_L^{-1/6} \to \infty$ and $R_0 \to \infty$. We find these latter results to be intuitively reasonable. The approximation works best when $\Psi$ is small. We find the approximate result $R_0 = \gamma x_E$ where $\gamma \approx 1.2 - 1.5$ with the given definition of parameters. This gives $x_E = (4/C_L)^{1/6}/(\gamma - 1)$ and $R_0 = (4/C_L)^{1/6}\gamma/(\gamma - 1)$. An example of the modified model with a curved interface between the solar wind and the Earth magnetic field is shown in Fig. A2. The ideal or desired result would have been a parabolic form for the magnetopause. The method of images is, however, not well developed for such problems. We can postulate a solution with a parabolic shape, where the curvature at the stagnation point is given through the foregoing analysis.

An impulse in the solar wind, be it in velocity or density or both, will give rise to a reduction in the distance $R_0 - x_E$, but at the same time it will induce also a change in the curvature $R_0$ of the part of the magnetosphere facing the sun. Within the present model there is no anisotropy in this curvature: it is the same in the plane parallel and perpendicular to the direction of the Earths magnetic dipole. The modified model can not account for the formation of the magnetotail.

## Appendix B: Currents induced in the ionosphere

The space-time varying electric and magnetic fields generated by the dynamic variations in the position and intensity of the Chapman-Ferraro current system induces currents in the Earth's near ionosphere. The ionosphere has a significant altitude variation in the Pedersen and Hall resistivities as well as in the magnetic field aligned conductivity. The problem can be solved only by considering strongly idealized conditions, but these can be helpful by giving insight into some general features. For conditions with large plasma parameters, we have a high plasma conductivity $\xi$, but it will never be super-conducting

conditions, so it will be penetrated by a steady magnetic field. For dynamic conditions with large magnetic Reynolds number $U\mathcal{L}/\xi$ where $\mathcal{L}$ is a characteristic scale size and $U$ some characteristic velocity, we can assume that the ionosphere acts passively for time-stationary magnetic conditions, but responds as an ideally conducting "shell" to rapid temporal changes in electric and magnetic fields (Davidson, 2001; Pécseli, 2012). This limit has an exact analytical solution when we assume that the moving image dipole field imposes a locally homogeneous time varying magnetic field at the Earth. In this case we can formulate the question as: "what secondary image dipole is needed to make the boundary conditions at the conducting shell to be fulfilled?", the boundary condition being that the normal component of the magnetic field vanishes at the conducting shell. For the simple limit mentioned before the answer is readily found. We let $B_I(t) = \mu_0 Q_I / 2\pi (2R(t))^3$ be the locally homogeneous magnetic field originating from the moving image dipole at a distance of $2R(t)$ from the Earth, see Figs. 1 and 3. We now introduce one more image magnetic dipole with dipole moment $\mathbf{Q}_S$ placed at the Earth's center. For the radial and angular variations of the total magnetic field we have

$$B_r = \left( B_I - \frac{\mu_0 Q_S}{2\pi r^3} \right) \sin\lambda \tag{B1}$$

$$B_\theta = \left( B_I + \frac{\mu_0 Q_S}{2\pi r^3} \right) \cos\lambda, \tag{B2}$$

where it is also here most convenient to measure the angle $\lambda$ from the equator of the dipole. With the given choice of polarities we find from (B1) that the normal component of the magnetic field at the conducting shell with radius $R_E$ vanishes for $Q_S(t) = 2\pi B_I(t) R_E^3 / \mu_0$. At $r = R_E$ we then find from (B2) the angular magnetic field component $B_\theta(t) = 2B_I(t)\cos\lambda$. The corresponding surface current density at the bottom of the ionosphere is then $K_S(t) = 2(B_I(t)/\mu_0)\cos\lambda$ in the direction perpendicular to $\mathbf{Q}_S$ in the azimuthal direction $\perp \mathbf{B}$, thus contributing to the electrojet current. From Fig. 4 we note that the assumption of a locally homogeneous magnetic field imposed by the image dipole representing the Chapman-Ferraro current system can be questioned when the magnetosphere is strongly compressed. In such a case we can obtain a slight improvement of the previous result by displacing the image dipole $\mathbf{Q}_S$ slightly in the sun-ward direction. An illustrative result is shown in Fig. B1. An ideally conducting ionosphere would thus shield ground stations completely from temporal variations of the magnetic field. It seems a safe conclusion that a *partially* conducting ionosphere will reduce the effects of the electric and magnetic field variations as detected on ground. The salty waters of the oceans also act as a conductor, albeit poor in comparison to the ionosphere. The time varying electric fields will induce currents also in the oceans, and the resulting (weak) magnetic field variations might be detectable by ground stations.

The time variation of the magnetic field at $r > R_E$ follows the variation in $B_I(t)$ directly within the given model, see Fig. 8.

### Appendix C: Dimensional arguments

Some results can be derived from simple dimensional arguments (Buckingham, 1914). Consider for instance the distance between the Earth and the magnetopause. The basic parameters are here the solar wind ram-pressure $n\overline{M}U^2$ where we note that the solar wind density and velocity will always appear in this combination so there is no generality gained by taking the

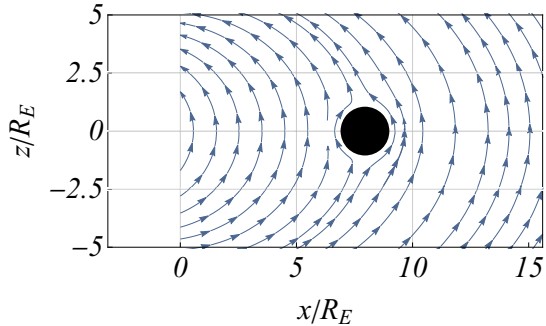

**Figure B1.** *Illustration of the change in the magnetic field in the vicinity of the Earth in response to a change in the Chapman-Ferraro current assuming an ideally conducting ionosphere. The figure shows only the time-varying part of the magnetic field: the Earth's steady state magnetic field is not included.*

variables separately. Similarly, $\mu_0$ and $Q$ will also appear together, but since $\mu_0$ also enters the magnetic pressure it has to be included explicitly as well. The mass loading will be here $\rho D$. The dimension matrix for the problem is given by the table

|   | $n\overline{M}U^2$ | $R$ | $\mu_0 Q$ | $\mu_0$ | $\rho D$ | $T_0$ |
|---|---|---|---|---|---|---|
| $T$ | -2 | 0 | -2 | -2 | 0 | 1 |
| $L$ | -1 | 1 | 3 | 1 | -2 | 0 |
| $M$ | 1 | 0 | 1 | 1 | 1 | 0 |
| $A$ | 0 | 0 | -1 | -2 | 0 | 0 |

For a time stationary problem where the magnetopause is at rest we have time $T_0$ in the sixth column to vanish from the problem, and similarly the inertia term $\rho D$ can not have any effect either. Then the first and third rows are proportional. We write from left to right in terms of the variables on the top in the dimension matrix

$$\left(\frac{M}{LT^2}\right)^{\alpha_1} \times L^{\alpha_2} \times \left(\frac{L^3 M}{T^2 A}\right)^{\alpha_3} \times \left(\frac{LM}{T^2 A^2}\right)^{\alpha_4},$$

and determine the exponents $\alpha_j$ is such a way that the exponents of *mass*, of *time*, of *length* and of *current* are each equal to zero. Evidently this requires $\alpha_1 = \alpha_4$, $\alpha_2 = 6\alpha_4$ and $\alpha_3 = -2\alpha_4$. We arrive at the combination of parameters

$$\left(\frac{n\overline{M}U^2 R^6 \mu_0}{(\mu_0 Q)^2}\right)^{\alpha_4} = 1. \tag{C1}$$

Choosing $\alpha_4 = 1$ we arrive at the result found in (1), apart from a numerical constant that can not be recovered by dimensional analysis. Given the parameters entering the combination in (1) is thus the only possible one (except a numerical constant) for the stationary problem with the given assumptions. Experimentally verifiable deviations from the scaling will thus indicate that there are missing parameters in (1). One possibility could be the solar wind resistivity: the magnetic Reynolds number there (Davidson, 2001; Pécseli, 2012) is large but still finite so the assumption of ideal conductivity in the application of the method of images can be challenged. We believe that a systematic investigation of this problem is worthwhile.

The dynamic problem is somewhat more complicated. Here we retain also the two last columns in the dimension matrix, and note that any dimensionally correct combination of parameters can be multiplied by the left side of e.g. (C1), or by $(n\overline{M}U^2T_0^2R^{-1}(\rho D)^{-1})^{\alpha_0}$ to an arbitrary power $\alpha_0$. We can thus decide that some parameters are kept constant, and determine the dimensionally correct combination of the rest. To derive a characteristic period of oscillation $T_0$ we first note that by Newton's second law $\rho D d^2\Delta/dt^2 = Force$ for the displacement $\Delta$ of the magnetopause, we expect the product $\rho D T_0^{-2}$ to appear, rather than these quantities individually. As long as the solar wind pressure is kept constant the variation of the force with varying displacement $\Delta$ will be due to the variations of the magnetic pressure with varying distance. We ignore the first column. From the dimension matrix we then have

$$L^{\alpha_1} \times \left(\frac{L^3M}{T^2A}\right)^{\alpha_2} \times \left(\frac{LM}{T^2A^2}\right)^{\alpha_3} \times \left(\frac{M}{L^2T^2}\right)^{\alpha_4}.$$

We find $\alpha_1 = 7\alpha_4$, $\alpha_2 = -2\alpha_4$, $\alpha_3 = \alpha_4$, giving

$$\left(R^7 \frac{\mu_0}{\mu_0^2 Q^2} \frac{\rho D}{T_0^2}\right)^{\alpha_4} = 1.$$

Taking again $\alpha_4 = 1$, this result is consistent with (3) apart from a numerical factor.

A damping factor arises by a phase difference between the magnetopause displacement $\Delta$ and the velocity $d\Delta/dt$, where it is taken into account that it is the relative velocity between the solar wind and $d\Delta/dt$ that matters. A dimensional analysis of this problem will be lengthy.

## Appendix D: Radiation belt details

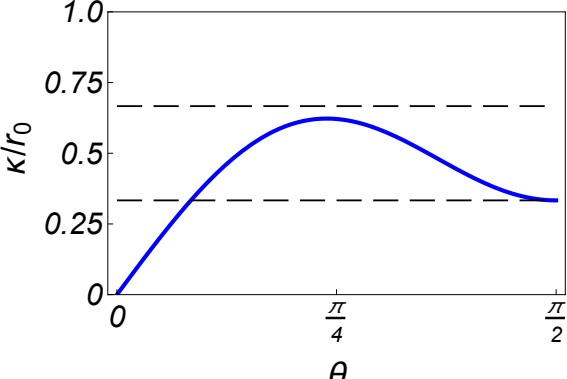

**Figure D1.** *The variation of the normalized radius of curvature $\kappa/r_0$ found as we follow a magnetic field-line specified by $r_0$, which is the maximum distance it reaches from the Earth's center. Two thin dashed lines give $\kappa/r_0 = 1/3$ and $\kappa/r_0 = 2/3$ for reference. Here $\theta$ measures the angle from the magnetic pole.*

In this appendix we summarize some details concerning the radiation belt heating due to the asymmetric compression caused by the motion of the Chapman-Ferraro current system. We model the basic averaged gyrocenter velocities by the $\nabla B$-drifts

$$\overline{\mathbf{U}}_{\nabla B} = -\frac{MU_\perp^2}{2q}\frac{\nabla B \times \mathbf{B}}{B^2} \sim \frac{\mu}{q\kappa}, \tag{D1}$$

and curvature drifts (Chen, 2016)

$$\overline{\mathbf{U}}_{cu} = \frac{MU_\parallel^2}{q}\frac{\boldsymbol{\kappa} \times \mathbf{B}}{B^2\kappa^2} \sim 2\left(\frac{U_\parallel}{U_\perp}\right)^2\frac{\mu}{q\kappa}, \tag{D2}$$

where $\kappa$ is the radius of curvature for the magnetic field line, introducing also the magnetic moment $\mu = \frac{1}{2}MU_\perp^2/B$. With $\nabla B$ and $\boldsymbol{\kappa}$ having opposite directions, the two velocities $\overline{\mathbf{U}}_{\nabla B}$ and $\overline{\mathbf{U}}_{cu}$ add-up (Chen, 2016). Using a magnetic dipole as an approximation we have the expression for a magnetic field line in spherical coordinates as $r = r_0\sin^2\theta$, where $r_0$ specifies the reference position on the selected magnetic field line as the distance from the dipole center measured at magnetic equator, $\theta = \pi/2$. The radius of curvature can then be found by standard expressions (Pécseli, 2012) as illustrated in Fig. D1. The figure shows the range of validity of (D2) if we assume $\kappa$ to be constant.

The time $T$ to circle Earth with the combination of the gradient drift and curvature drifts depends on the selected radius and the particle energy. For a 1 MeV particle at a distance of $5R_E$ it takes approximately $10^3$ s, or $\sim 15$ minutes. Combining the velocities in (D1) and (D2) to $\overline{U}$ we have $T = 2\pi r/\overline{U} \sim 1/(r\mathcal{W})$ at some distance $r$ from the magnetic dipole center in the magnetic equator plane with $\mathcal{W}$ being the particle energy. Similarly we have the scaling $\overline{U} \sim r^2\mathcal{W}$, implying a particle dispersion in the sense that energetic particles arrive first, see Figs. 22 and 23.