# Peer review of "Impulse-driven oscillations of the near-Earth's magnetosphere"

_EGUsphere, 2022_

## Referee Comment (RC2)

Title: Impulse-driven oscillations of the near-Earth's magnetosphere
Author(s): Hiroatsu Sato et al.
MS No.: egusphere-2022-232

[Comments]
There are major questions as listed below.
1. The ground magnetic variations at high latitudes are described as being caused by magnetopause currents. However, ground magnetic variations at high latitudes are caused by ionospheric Hall currents driven by the electric field created by the dynamo in the outer magnetosphere (e.g., Tanaka et al., 2020 JGR, https://doi.org/10.1029/2019JA027172). Furthermore, the field-aligned currents generated by the dynamo flow to the global ionosphere via the polar ionosphere, resulting in simultaneous occurrence of the preliminary impulse (PI) and main impulse (MI) of SC at high latitude and equator (Araki, 1994 AGU book). The authors are recommended to discuss their results in the context of the current system between the magnetosphere and ionosphere. The dynamos of the PI and MI have also been reproduced by the global simulations (Slinker et al., 1999 JGR; Fujita et al, 2003 JGR).
2. Damping of the magnetosphere is attributed to the non-linearity of the equation (4). On the other hand, the FACs flow into the polar ionosphere and further to the global ionosphere, where the energy is consumed by the Pedersen currents (e.g., Kikuchi et al., 2021 EPS). When discussing the damping of the ground magnetic fields, it is advisable to discuss the energy loss in the ionosphere.
3. The motion of plasma is described as being earthward, but the calculated and observed electric fields presented in the present paper are directed from the dawn to dusk, which drives sunward motion of plasma. Explanation or comments are required for the difference between the compression of the magnetopause and sunward motion of the magnetospheric plasma.

[Others]
Line 116
The Faraday's law is equivalent to E=UxB?

Line 124
In Figure 9, the electric field is from the dawn to dusk, which drives sunward motion of plasma in the magnetosphere. If the electric field is induced by the increase in Bz and carried by the compressional MHD wave toward the Earth, the direction of the electric field must be westward, i.e., from the dusk to dawn. How is the dawn-to-dusk electric field generated by the moving magnetopause currents?

Line 147
Draw current vectors on the current lines of the FACs. Previous studies using the global simulations have shown that two kinds of FACs are generated by the compression of the

magnetosphere (Slinker et al., 1999 JGR; Fujita et al., 2003 JGR; Tanaka, 2007 SSR), supplying the electric field and currents of the PI and MI of SC. The FAC pair inside the outside pair is also produced by the magnetopause currents?

Line 149
Please specify the energies of radiation belt particles and of particles that work as a generator of the FAC.

Line 157
   Please note that the infinite inner impedance does not allow electric currents to flow, since I=V/(r+R) where r and R are the internal and load resistivities.

Line 160
Please note that the FAC is generated by the high-pressure plasma so that the pressure gradient force balances JxB force of the dynamo current J (Tanaka, 2007 SSR).

Lines 160-163
The equations for the current are the same in the warm and cold plasma regions. Please comment on the difference between the nature of the two currents.

Line 209
Ground magnetic disturbances are caused by ionospheric currents, particularly at auroral and subauroral latitudes (e.g., Araki et al., 1997 JGR). At middle and low latitudes, the magnetic fields are caused by magnetopause currents superimposed by weak ionospheric currents. At the dayside equator, the ionospheric Cowling currents work as a major source for the equatorial SC (Araki, 1994 AGU book).

Line 235
Figure 17 shows a typical SC in the morning, composed of positive/negative PI and negative/positive MI at lower/higher latitude part of the IMAGE magnetometer array. These magnetic fields are caused by ionospheric Hall currents surrounding the FACs (e.g., Kikuchi et al., 2022 Frontiers, doi: 10.3389/fspas.2022.879314). Note that the onset of PI at higher latitude (NAL, BJN) is simultaneous with those at lower latitude (NUR,,,,), because the ionospheric currents flow at the speed of light to the global ionosphere, including the equator (Kikuchi et al., 2021 EPS, DOI: 10.1186/s40623-020-01350-8). Magnetopause currents cause magnetic fields at low latitude, which is DL according to Araki (1994) model.

Line 248
Which part of the data is believed to have been caused by FAC?

Line 253

The electric fields measured by the satellites are from the dawn to dusk (Fig. 21), same as in the model calculation (Fig.9). The ExB drift velocity is sunward, which is opposite to the earthward motion of the magnetopause. The electric field observed in the ionosphere at middle latitude is also directed from the dawn to dusk, which lifts up the dayside ionosphere (Kikuchi et al., 2016 JGR, doi:10.1002/2015JA022166).

Line 259
RBSP spacecrafts are located deep inside the magnetosphere, not close to the magnetopause. The location of spacecrafts should be explicitly mentioned in the discussion.

Line 300
Phase relationships among the ground magnetic variations should be mentioned. If the magnetic variations are caused solely by the magnetopause currents, we would see coherent variations in multiple locations.

---

## Author Comment (AC1)

*The referee's comments are reproduced below, with authors response (in italics) to the separate points. Corrections in the manuscript will be marked in red. Deletions are not specified. The revised manuscript, with correction in colour, will be made open when both sets of referee comments have been taken into account.*

The paper builds on and analyzes the performance of a simple model of solar wind – Earth's magnetic field interaction. Consequences of a sudden pressure pulse in the solar wind for the dynamics of the system are then discussed and the respective variations are qualitatively evaluated. These are, in turn, compared with ground-based magnetometer and Van Allen Probes measurements. The claimed reasonable agreement is interpreted in terms of this simple model being, to the lowest order, sufficient to model the near-Earth magnetosphere.

I find the paper rather interesting, as such simple-model approach is quite rarely seen nowadays. On the other hand (or perhaps because of that), I have some doubts/questions concerning the model formulation and its comparison with the measurements.

Detailed comments

Static limit in equation (1) and around: I feel this argumentation based on the pressure balance is well known. It would be more usual to have the solar wind dynamic pressure units (Figure 3) in nPa and to have the equation (1) in SI units. Also, it is worth noting that the -1/6 scaling resulting from this simple picture is often slightly violated in empirical magnetopause models, so I have some doubts about that "generally accepted" formulation.

*Authors response: yes, the basic scaling is well known as also stated in the paper. It can be derived from dimensional reasoning (see Appendix B) but the numerical coefficient is found empirically in other studies. The present simple model gives it analytically, we believe this is new, although this is not a major new result. The referee's remarks that the 1/6 scaling is violated by observations. Such a result will be dimensionally wrong and sounds strange if the deviations are significant. We would appreciate more information from the referee. As shown in Appendix B, the 1/6 result is the only one that is dimensionally correct with the given parameters, so if this law is violated there must be parameters missing: this would be an important observation even if we can not specify what is missing. Comments on this are inserted in the manuscript. We remade figure 3 to have the horizontal axis in nPa. Equation (1) is actually in SI-unuts and this is made more clear now.*

Equation (2) governing the assumed magnetopause oscillations: I believe that this is quite essential for the model formulation and should be better discussed and justified. First, what is the source of the inertia here? What typical values are found/considered? Do the typical speeds of magnetopause obtained here correspond to the observations? (these can be determined experimentally using multi-spacecraft measurements, Cluster was used for that as far as I know). Second, the damping coefficient should be discussed better. It is said that it does not correspond to the dissipation, but is rather a result of the phase-lag in the mathematical formulation. Ok; but I would be hesitant to call this a "physical mechanism" – and the energy should perhaps still go somewhere (?)

*Authors response: The damping is caused by the asymmetry in the solar wind pressure: when the magnetopause is approaching (i.e., moving away from the Earth) the magnetopause is doing work on the solar wind, while in the receding phase it is opposite. The two cases are not symmetric since the ram solar wind pressure depends on the relative velocity between the solar wind and the magnetopause. In the approaching phase this force is large, while it is smaller in the receding phase. The work done in the two oscillation phases is different. The net result is a loss of energy from the oscillation. The initial transient time interval is different: here the solar wind pulse or shock arrives at*

*an interface at rest, and the oscillations are initiated to reach full amplitude. This explanation is now inserted in the text. The magnetopause inertia is discussed in more detail and references to observation and simulations are given.*

125-135: People typically consider ExB drift to be negligible for the radiation belts particles, as for high energies grad-B and curvature drifts dominate. I have thus some doubts about the calculation here. How was Figure 10 obtained? For what energies? What pitch angles? The asymmetry of magnetic field should result in some drift-shell splitting. None of this is discussed/described (and considered?).

*Authors response: we included the ExB drift of magnetic field lines (in an MHD sense) in the radial direction: Yes, perpendicular to B (evidently), the gradient B and curvature drifts are the important ones. We had a short discussion in Appendix C. In terms of MHD plasma dynamics we have the magnetic field lines moving with the ExB-drifts, the particles move with their respective dynamics on these magnetic field lines. The text is clarified a little more on this.*

160-165: what are the assumed values of the density here? The relative densities of high-energetic particles will be comparatively very low. Also, the energization of the radiation belt particles is typically due to (inward) radial diffusion, which, in turns, decreases the azimuthal drift velocity.

*Authors response: the mentioned decrease in azimuthal velocity was discussed in some detail in Appendix C, but is now emphasized in the text. Due to the compression of the sunward part of the Earth's magnetic field, the estimate of charged particle velocities based on a magnetic dipolar field will only serve as a guideline, but in principle we agree with the referee. The text is improved concerning the density of heated particles, but all we can state here is that the fraction is low, as argued by the referee.*

Comparison with observations: it remains quite unclear what the model can or cannot predict and how this match or does not match the observations. The sudden change of the magnetic field measured due to the increase of the Chapman-Ferraro current (and magnetopause moving to lower distances) at the time of the pressure pulse is well known. The model might be in principle able to predict the subsequent oscillation period (?) and attenuation of the magnetic field pulsations (?), but these are difficult to see in the data and some more elaborated comparison with the model output is missing. Instead, the shock parameters (not really too relevant for the model evaluation (?)) are described.

*Authors response: we believe it is important to quantify the two shocks (the differences are significant), their relative strengths in particular, so some effort was made to derive and present these data. The Conclusion section contains a summary of features not covered by the present model.*

There was recently quite a large number of papers dealing with the shock effects on radiation belts / magnetospheric plasma waves which seem to be quite ignored in the present manuscript (e.g., Sun et al. (2015), doi: 12014JA020754; Foster et al. (2015), doi: doi:10.1002/2014JA020642; Tsuji et al. (2017), doi: 10.1002/2016JA023704; Blum et al. (2021), doi: 10.1029/2021GL092700 – and most likely many others they cite/are cited by).

*Authors response: unfortunately, the referee is quite right here: these references were missing and we have no objections to including them, in particular also because they are quite recent. On the other hand, they are not essential for our analysis. The DOI-number for Sun et al (2015) was incomplete, but hopefully we found the correct reference. The amount of literature on the subject is vast and the main contribution of our work may be the simplicity, yet usefulness, of our model.*

295: This configuration of the three dipoles should be better described already in the beginning, not just here in the conclusions. The claimed "good agreement" between the model and observations is not really demonstrated.

*Authors response: in principle the referee is of course right, but we should emphasize that in reality there are no objective measures for good agreement, only for no agreement. Given the simplicity of our model we find that it is able to predict surprisingly many features of the geomagnetic disturbances induced by solar-wind shocks.*

---

## Author Comment (AC2)

We have received two referee reports on our paper and write two separate responses. The 2. referee report is reproduced below and authors responses are written in *italics* after each point. The revised version of the paper shows additions and nontrivial changes in red color. Deletions are not marked. Due to the automatic LaTex formatting, we could not color new references but there are several, as seen when comparing the present and original versions. The revised paper with colors will be made public when we have included changes due to both referee comments in it.

*We have some general comments to this referee report. First of all, we find it constructive and with a clear helpful intent, but in our opinion it fails in identifying the scope of our analysis. The starting point in our analysis is a simple model for estimating the "stand-off" distance between the Earth and the magnetopause. The model has no free parameters and supported by a dimensional analysis. The basics of the static model are known. Our question was then to what extent this static model could be generalized to include also dynamic effects? It is very easy to find shortcomings of such simple models, but our question was (as also stated in the paper) to see what it does right. In our conclusion we had and have a list of the shortcomings of the model: its ability to give an account of field aligned currents was mentioned as one. Some elements of such currents can be argued, but a complete description is impossible within this model. We have, however, expanded on the relations to other works, Araki 1994 in particular, this was one of the suggestions of the referee.*

Title: Impulse-driven oscillations of the near-Earth's magnetosphere

Author(s): Hiroatsu Sato et al.

MS No.: egusphere-2022-232

[Comments]

There are major questions as listed below.

The ground magnetic variations at high latitudes are described as being caused by magnetopause currents. However, ground magnetic variations at high latitudes are caused by ionospheric Hall currents driven by the electric field created by the dynamo in the outer magnetosphere (e.g., Tanaka et al., 2020 JGR, https:// doi.org/10.1029/2019JA027172). Furthermore, the field-aligned currents generated by the dynamo flow to the global ionosphere via the polar ionosphere, resulting in simultaneous occurrence of the preliminary impulse (PI) and main impulse (MI) of SC at high latitude and equator (Araki, 1994 AGU book). The authors are recommended to discuss their results in the context of the current system between the magnetosphere and ionosphere. The dynamos of the PI and MI have also been reproduced by the global simulations (Slinker et al., 1999 JGR; Fujita et al, 2003 JGR).

Damping of the magnetosphere is attributed to the non-linearity of the equation (4). On the other hand, the FACs flow into the polar ionosphere and further to the global ionosphere, where the energy is consumed by the Pedersen currents (e.g., Kikuchi et al., 2021 EPS). When discussing the damping of the ground magnetic fields, it is advisable to discuss the energy loss in the ionosphere.

*Authors response: if eq. (4) is linearized, it still contains a damping, so it is not a nonlinear effect. The damping is caused by the asymmetry in the solar wind pressure: when the magnetopause is approaching (i.e., moving away from the Earth) the magnetopause is doing work on the solar wind, while in the receding phase it is opposite. The two cases are not symmetric since the solar wind ram-pressure depends on the relative velocity between the solar wind and the magnetopause. In the*

*approaching phase this force is large, while it is smaller in the receding phase. The work done in the two oscillation phases is different. Integrated over an oscillation period 2\pi/\Omega, the oscillations loose net energy to the solar wind so the net result is a damping of the oscillations. The initial transient time interval is different: here the solar wind pulse or shock arrives at an interface at rest, and the oscillations are initiated to reach full amplitude. We have no objections to cite the work by Kikuchi et al (hoping we found the correct reference) but we see no reason why this type of damping should be superior to the one we found.*

The motion of plasma is described as being earthward, but the calculated and observed electric fields presented in the present paper are directed from the dawn to dusk, which drives sunward motion of plasma. Explanation or comments are required for the difference between the compression of the magnetopause and sunward motion of the magnetospheric plasma.

*Authors response: we imagine this comment refers to a plotting error in figure 9 as also mentioned later. The error is corrected now. We thank the referee for noting this.*

[Others]

Line 116

The Faraday's law is equivalent to E=UxB?

*Authors response: That part of the sentence has been deleted.*

Line 124

In Figure 9, the electric field is from the dawn to dusk, which drives sunward motion of plasma in the magnetosphere. If the electric field is induced by the increase in Bz and carried by the compressional MHD wave toward the Earth, the direction of the electric field must be westward, i.e., from the dusk to dawn. How is the dawn-to-dusk electric field generated by the moving magnetopause currents?

*Authors response: the referee is right, we made a mistake in figure 9, the arrows should have been with opposite polarity. The error is corrected. Thank you for pointing this out!*

Line 147

Draw current vectors on the current lines of the FACs. Previous studies using the global simulations have shown that two kinds of FACs are generated by the compression of the magnetosphere (Slinker et al., 1999 JGR; Fujita et al., 2003 JGR; Tanaka, 2007 SSR), supplying the electric field and currents of the PI and MI of SC. The FAC pair inside the outside pair is also produced by the magnetopause currents?

*Authors response: our original argument was based a dissimilarity of the heated and unheated parts of the radiation-belt plasmas. The original text was admittedly schematic and a more detailed calculations show that "our" effect is quite small. It has no meaning to retain discussions of a small effect when the rest of the paper deals with large-scale bulk plasma phenomena, so this particular discussion is now deleted. Instead we introduce a reference (with a short discussion) to Araki's 1994*

*results, the polarization current in particular, as advocated by the referee. The figure is completed with an arrow, as suggested.*

Line 149

Please specify the energies of radiation belt particles and of particles that work as a generator of the FAC.

*Authors response: this discussion is deleted for reasons mentioned before.*

 Line 157

 Please note that the infinite inner impedance does not allow electric currents to flow, since I=V/(r+R) where r and R are the internal and load resistivities.

*Authors response: there seems to be a misunderstanding here. The current generators are circuit elements having an infinite internal resistance. Current generators force a current in the same way as a voltage generator impose a voltage difference. The volage generator has a vanishing internal resistance, but it does not short circuit the current. Given an ideal current generator, the resulting voltage differences are a consequence of the load combined with the forced current. Given an ideal voltage generator, the currents flowing are a consequence of the load and the imposed voltage. In the ideal limit a current generator forces a current into an infinite resistance, giving infinite voltage differences. An ideal voltage generator on the other hand forces a potential between two short-circuited points, giving infinite currents. These extreme cases are remedied by finite internal resistances and the two generators are related by Thevenin's and Norton's theorems. For instance, polarization currents are induced by imposing a cross-field velocity, not by a potential difference.*

 Line 160

Please note that the FAC is generated by the high-pressure plasma so that the pressure gradient force balances JxB force of the dynamo current J (Tanaka, 2007 SSR).

*Authors response: this is so for steady state conditions where we have force balance. The present study deals with transient phenomena.*

 Lines 160-163

The equations for the current are the same in the warm and cold plasma regions. Please comment on the difference between the nature of the two currents.

*Authors response: as mentioned elsewhere, that discussion is now deleted.*

 Line 209

Ground magnetic disturbances are caused by ionospheric currents, particularly at auroral and subauroral latitudes (e.g., Araki et al., 1997 JGR). At middle and low latitudes, the magnetic fields are caused by magnetopause currents superimposed by weak ionospheric currents. At the dayside equator, the ionospheric Cowling currents work as a major source for the equatorial SC (Araki, 1994 AGU book).

*Authors response: in general, we agree with the referee on this, and actually the text contained such statements, but we now gave the point greater emphasis.*

Line 235

Figure 17 shows a typical SC in the morning, composed of positive/negative PI and negative/positive MI at lower/higher latitude part of the IMAGE magnetometer array. These magnetic fields are caused by ionospheric Hall currents surrounding the FACs (e.g., Kikuchi et al., 2022 Frontiers, doi: 10.3389/fspas.2022.879314). Note that the onset of PI at higher latitude (NAL, BJN) is simultaneous with those at lower latitude (NUR,,,,), because the ionospheric currents flow at the speed of light to the global ionosphere, including the equator (Kikuchi et al., 2021 EPS, DOI: 10.1186/s40623-020-01350-8). Magnetopause currents cause magnetic fields at low latitude, which is DL according to Araki (1994) model.

*Authors response: we have no objections to referencing the work of Araki (1994) and added this together with a short discussion. The bipolar signal seen in our Fig. 17 at auroral latitudes corresponds to the DP-type perturbations described by Araki (1994). The interpretation of this signature given in there (auroral zone, morning local time) is in terms of an M-I-coupling illustrated in Fig. 12 in that work. The signatures shown in the present work at Svalbard latitudes are explained in terms of lobe-cell polar cap convection with an associated Hall-current.*

*In the discussion of the Araki-model of M-I current system in relation to our manuscript we might add the following. This is related to the bipolar DP-type perturbation (at auroral latitudes). (DL is Arakis term for perturbations in the equatorial region). In the M-I system of Araki (his Fig. 12) the auroral ionosphere is coupled (via FACs) to his Jp-current in the magnetosphere (directed dusk to dawn), which is located well inside the magnetopause. Jp is a polarization current (giving rise to FAC-system). Our figure 2 gives the dawn-dusk directed Chapman-Ferraro current (which is moving inward during the external pressure enhancement*

*We point out that Araki (1994) splits the the SC magnetic perturbation in two components (called DP and DL), corresponding to two different sources (as described in his Figs. 11 and 12): i) the CF-current (as illustrated in our Fig. 2), giving rise to Arakis DL-type, similar to that shown in our Fig. 1, and ii) the polarization current of the Araki-model (Jp) with associated FAC-currents and currents in the auroral ionosphere, accounting for the DP-type perturbation (illustrated by the bipolar signature in our Fig.17). In addition, we observe at the highest latitudes (at Svalbard stations in Figure 17) a perturbation that may be attributed to lobe cell convection (and associated Hall currents) under the prevailing strongly northward IMF conditions.*

Line 248

Which part of the data is believed to have been caused by FAC?

*Authors response: as mentioned, the part referring to the polar regions.*

Line 253

The electric fields measured by the satellites are from the dawn to dusk (Fig. 21), same as in the model calculation (Fig.9). The ExB drift velocity is sunward, which is opposite to the earthward motion of the magnetopause. The electric field observed in the ionosphere at middle latitude is also directed from the dawn to dusk, which lifts up the dayside ionosphere (Kikuchi et al., 2016 JGR, doi:10.1002/2015JA022166).

*Authors response: as mentioned elsewhere, we had a plotting error in our original figure 9. The y-component of Electric field components shown from the van Allen probes is in the opposite direction. The model electric field is obtained for electromagnetic variations in vacuum, while the satellites are in the radiation belts so our comparison is in reality misplaced if it is applied in detail, but we still find the oscillation period detected by the satellites interesting since it agrees with the model predictions. Our main interest on the electric field signal is found in its use as a reference "time-marker" for estimating the time delay for the arrival of the energetic particles.*

Line 259

RBSP spacecrafts are located deep inside the magnetosphere, not close to the magnetopause. The location of spacecrafts should be explicitly mentioned in the discussion.

*Authors response: the spacecraft position was given explicitly in a separate figure.*

Line 300

Phase relationships among the ground magnetic variations should be mentioned. If the magnetic variations are caused solely by the magnetopause currents, we would see coherent variations in multiple locations.

*Authors response: our figures 14 and 18 showing time variations for GUA, DLT and M08 stations were inserted for this reason, they show precisely as the referee argues, synchronous variations at widely separated locations. The available journal space does not allow detailed documentation, but we insert below as an illustration for one of our events using stations uniformly scattered over the globe in a band along equator. Note that damped oscillations following the arrival of the shock are noticeable in most signals.*

Magnetometer Data IPM

N
E
Z

nT

Magnetometer Data GUA

N
E
Z

nT

Magnetometer Data PPT

N
E
Z

nT

Magnetometer Data M08

N
E
Z

nT

Magnetometer Data DLT

N
E
Z

nT

Magnetometer Data JAI

N
E
Z

nT

2015/03/17 04:20 UTC   2015/03/17 04:38 UTC   2015/03/17 04:56 UTC   2015/03/17 05:14 UTC   2015/03/17 05:32 UTC   2015/03/17 05:50 UTC

---

## Author Response (AR2)

Authors response to referee comments/remarks. The comments are taken separately as listed below, with authors responses in *italics*. In the revision of the paper, we have marked the relevant additions in blue. The previous additions are still there; these are marked with red text.

• Does there exist (or if you have carried some) comparisons with experimentally determined magnetopause speeds? What are the typical model magnetopause speeds?

*Authors response: Some observations are now reported concerning magnetopause speeds and we have added a few comments on this,* i.e., *Phan, T. D. and Paschmann, G.: Low-latitude dayside magnetopause and boundary layer for high magnetic shear: 1. Structure and motion, J.Geophys. Res. Space Phys., 101, 7801–7815, doi:10.1029/95JA03752, 1996. and Paschmann, G. et al.: Structure of the dayside magnetopause for low magnetic shear, J. Geophys. Res. Space Phys., 98, 13 409–13 422, doi:10.1029/93JA00646, 1993. The observations of this type have a common problem due to the accelerated motion of the magnetopause. The observed velocities are merely samples in this time-interval. We point this out in the text. The observed orders of magnitudes are consistent with our results.*

• Why did you prefer to base your argumentation on ExB drift? Perhaps a direct explanation of why the velocity of compression is not important (particularly because the total magnetic field in this model is the sum of the Earth's magnetic field and the magnetic field resulting from the Chapman-Ferraro current) would be more appropriate?

*Authors response: the text already had a comment on the ExB-drifts: they are the magnetic field line velocities (in an MHD sense as stated; generally, it is meaningless to have magnetic field lines moving, they are merely a guide to be drawn at fixed times. However, in MHD we can let magnetic lines move without introducing inconsistencies. This is mentioned in most textbooks.) We use this as a basis or reference velocity, where in addition the individual particle motions are given by their respective drift velocities, i.e., gradient drifts, polarization drifts, etc. Concerning the compression velocity, the referee has a point. The text as it was written referred to adiabatic compression of an isolated system. We have added more details on this, arguing that the compression is fast (as also supported by the observed velocities of the magnetopause), so the contact with surroundings had no time to be materialized.*